# Neuronal migration depends on blood flow in the adult mammalian brain

Takashi Ogino[1†], Akari Saito[1†], Masato Sawada[1,2], Shoko Takemura[1], Yuzuki Hara[1], Kanami Yoshimura[1], Jiro Nagase[1], Honomi Kawase[1], Takamasa Sato[1], Hiroyuki Inada[3], Vicente Herranz-Pérez[4,5], Yoh-suke Mukouyama[6], Masatsugu Ema[7], José Manuel García-Verdugo[4], Junichi Nabekura[1,3], Kazunobu Sawamoto[1,2]*

[1]Department of Developmental and Regenerative Neurobiology, Nagoya City University Graduate School of Medical Sciences, Nagoya, Japan; [2]Division of Neural Development and Regeneration, National Institute for Physiological Sciences, Okazaki, Japan; [3]Division of Homeostatic Development, Department of Developmental Physiology, National Institute for Physiological Sciences, Okazaki, Japan; [4]Laboratory of Comparative Neurobiology, Cavanilles Institute, University of Valencia, Valencia, Spain; [5]Department of Cell Biology, Functional Biology and Physical Anthropology, University of Valencia, Burjassot, Spain; [6]Laboratory of Stem Cell and Neuro-Vascular Biology, Genetics and Developmental Biology Center, National Heart, Lung, and Blood Institute, National Institutes of Health, Bethesda, United States; [7]Department of Stem Cells and Human Disease Models, Research Center for Animal Life Science, Shiga University of Medical Science, Otsu, Japan

*For correspondence:
sawamoto@med.nagoya-cu.ac.jp

[†]These authors contributed equally to this work

Competing interest: The authors declare that no competing interests exist.

## eLife Assessment

This **fundamental** work provides novel insights into the blood flow-dependent mechanisms of neuronal migration and the role of Ghrelin signaling in the adult brain. The authors present **convincing** evidence that newborn rostral migratory stream (RMS) neurons are closely situated alongside blood vessels, preferentially along arterioles, and that migratory speed is correlated with blood flow. They also provide evidence (in vitro and some in vivo) that Ghrelin from blood is involved in augmenting RMS neuron migration speed.

**Abstract** In animal tissues, several cell types migrate along blood vessels, raising the possibility that blood flow influences cell migration. Here, we show that blood flow promotes the migration of new olfactory-bulb neurons in the adult mammalian brain. Neuronal migration is facilitated by blood flow, leading to accumulation of new neurons near blood vessels with abundant blood flow. Blood flow inhibition attenuates blood vessel-guided neuronal migration, suggesting that blood contains factors beneficial to neuronal migration. We found that ghrelin, which is increased in blood by hunger, directly influences neuronal migration. Ghrelin signaling promotes somal translocation by activating actin cytoskeleton contraction at the rear of the cell soma. New neurons mature in the olfactory bulb and contribute to the olfactory function for sensing odorants from food. Finally, we show that neuronal migration is increased by calorie restriction, and that ghrelin signaling is involved in the process. This study suggests that blood flow promotes neuronal migration through blood-derived ghrelin signaling in the adult brain, which could be one of the mechanisms that improves the olfactory function for food-seeking behavior during starvation.

## Introduction

Blood vessel-guided cell migration has been reported in various types of cells, including lymphatic endothelial cells in the embryonic zebrafish trunk (*Bussmann et al., 2010*), Schwann cells, oligoden-drocyte precursor cells, glioma cells, and astrocyte progenitors in rodent tissues (*Cattin et al., 2015*; *Farin et al., 2006*; *Tabata et al., 2022*; *Tsai et al., 2016*), suggesting that it is a common mechanism for efficient cell migration within complex animal tissues. Previous reports have extensively studied the role of blood vessels as a physical scaffold for migration of neurons produced in the postnatal brain tissues (*Bovetti et al., 2007*; *Fujioka et al., 2017*; *Grade et al., 2013*; *Ohab et al., 2006*; *Snapyan et al., 2009*; *Sun et al., 2015*; *Thored et al., 2007*; *Whitman et al., 2009*; *Yamashita et al., 2006*). New neurons generated in the postnatal ventricular–subventricular zone (V-SVZ) migrate to the olfactory bulb (OB) through the rostral migratory stream (RMS), forming elongated aggregates called chains (*Doetsch and Alvarez-Buylla, 1996*; *Lois et al., 1996*; *Lois and Alvarez-Buylla, 1994*). Within the OB, new neurons migrate individually and radially and terminate their migration in the granule cell layer (GCL) or glomerular layer (GL), where they are differentiated into GABAergic interneurons and incorporated into the OB neural circuit (*Bressan and Saghatelyan, 2020*; *Kaneko et al., 2017*). Previous studies have shown that both chain-forming and individually migrating neurons use blood vessels as physical scaffolds for their migration in the RMS and OB (*Bovetti et al., 2007*; *Snapyan et al., 2009*; *Whitman et al., 2009*). However, whether blood flow plays a role in blood vessel-guided cell migration has not been reported.

Ghrelin, a peripheral hormone produced in the stomach, is delivered to the brain through the bloodstream and accumulates in the brain parenchyma, especially at high levels in the OB (*Rhea et al., 2018*). The ghrelin concentration in blood has been reported to increase during fasting (*Toshinai et al., 2001*; *Tschöp et al., 2000*). In addition to its role in regulating metabolism (*Al Massadi et al., 2017*; *Chopin et al., 2012*; *Soleyman-Jahi et al., 2019*; *Stoyanova and Lutz, 2021*), ghrelin influences neurogenesis in the V-SVZ and subgranular zone of the hippocampal dentate gyrus (*Hornsby et al., 2016*; *Kent et al., 2015*; *Kim et al., 2015*; *Li et al., 2014*; *Li et al., 2013*; *Moon et al., 2009*). However, its role in blood vessel-guided neuronal migration has not been studied.

To elucidate the effects of blood flow on blood vessel-guided cell migration, we focused on neuronal migration along blood vessels in the adult RMS and OB. The results showed that new neurons migrate along blood vessels with high blood flow and that their migration is affected by changes in blood flow. We also found that ghrelin promotes neuronal migration through activation of actin cytoskeleton contraction at the rear of the cell soma, indicating that blood flow influences neuronal migration via ghrelin signaling. Furthermore, we found that calorie restriction promotes the migration of OB neurons, suggesting that the blood flow-dependent mechanism of neuronal migration could be part of a system to improve sensory function in response to physiological change in the body.

## Results

### New neurons migrate along blood vessels with abundant blood flow

Previous studies have revealed that V-SVZ-derived new neurons migrate along blood vessels in the RMS and GCL (*Bovetti et al., 2007*; *Snapyan et al., 2009*; *Whitman et al., 2009*). However, the neuron–vessel interactions along the entire migration route remain unknown. Therefore, we first studied blood vessel-guided neuronal migration in the RMS and OB using three-dimensional imaging in 6- to 12-week-old adult mice, which enables analysis of the in vivo spatial relationship between new neurons and blood vessels. We measured the distance from the soma of new neurons, labeled with green fluorescent protein (GFP)-encoding adenovirus, to the nearest blood vessel labeled with RITC-Dex-GMA. The results showed that GFP+ new neurons interact closely with blood vessels in the RMS, GCL, external plexiform layer, and GL (*Figure 1A–D*, *Figure 1—video 2*; *Figure 1—videos 1*). In particular, in the RMS and GL, the majority of cells were present within 5 µm of the inner surface of vessels (*Figure 1D*), suggesting that new neurons in these regions frequently use blood vessels as migration scaffolds. To visualize blood vessels, we also used *Flt1-DsRed* transgenic mice, in which vascular endothelial cells were specifically labeled with DsRed (*Matsumoto et al., 2012*). Using *Dcx-EGFP/Flt1-DsRed* double transgenic mice, we observed close spatial relationships between new neurons and blood vessels (*Figure 1—videos 3 and 4*). Transmission electron microscopy revealed direct attachment of migratory neurons, identified as cells with an elongated cell body, a dark

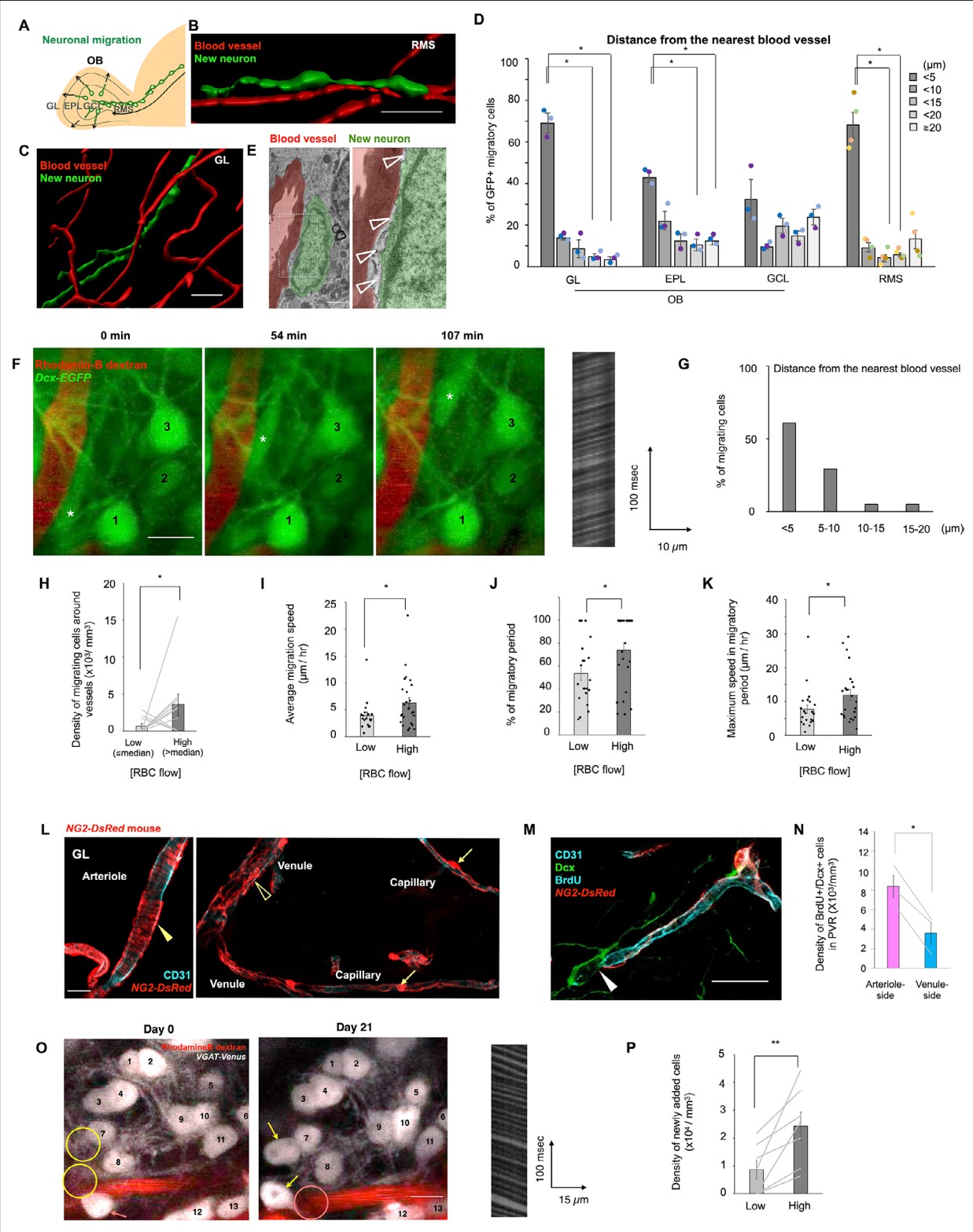

**Figure 1.** New neurons migrate along blood vessels with abundant flow in the adult brain. (**A**) Experimental scheme. (**B**) Three-dimensional reconstructed images of a new neuron (green) and blood vessels (red) in the rostral migratory stream (RMS) (**B**) and glomerular layer (GL) (**C**) of 6- to 12-week-old adult mice. (**D**) Distance between new neurons and nearest vessels in the olfactory bulb (OB) and RMS (one-way repeated measures ANOVA followed by Bonferroni's test; three and four mice for the analysis in the OB and RMS, respectively). (**E**) Transmission electron microscopy image of a new neuron (green) in close contact with a blood vessel (red) in the GL of a 6- to 12-week-old adult mouse. Astrocytes (clear arrowheads). (**F**) Time-

*Figure 1 continued on next page*

*Figure 1 continued*

lapse images of a migrating neuron (indicated by asterisks) in the GL of a 6- to 12-week-old *Dcx-EGFP* mouse. Red blood cell (RBC) flow is recorded as a two-photon line-scan image as shown in the right panel. Stationary cells are indicated by sequential numbers. (**G**) Average distance between migrating cells and nearest blood vessels (41 cells from 38 mice). (**H**) Density of perivascular migrating cells (Wilcoxon signed-rank test; 10 mice). (**I**) Average migration speed (Welch's *t*-test; low, 19 cells, high, 25 cells from 39 mice). (**J**) Percentage of migratory period (Mann–Whitney *U*-test; low, 19 cells, high, 24 cells from 38 mice). (**K**) Maximum migration speed (unpaired *t*-test; low, 22 cells, high, 24 cells from 39 mice). (**L**) Fluorescent images in the *NG2-DsRed* mouse GL. Arterioles, capillaries, and venules were characterized by band-like smooth muscle cells (solid arrowhead), pericytes (arrows), and fenestrated smooth muscle cells (clear arrowhead), respectively. CD31 (blue), DsRed (red). (**M**) Fluorescent image of a Dcx+/BrdU+ new neuron (solid arrowhead) attached to a capillary. Dcx (green), BrdU (blue), CD31 (blue, tube-like structures), DsRed (red). (**N**) Density of BrdU+/Dcx+ cells in the perivascular region of arteriole- and venule-side capillaries (paired *t*-test; three mice). (**O**) Two-photon images of GABAergic neurons (white) and a blood vessel (red) in the *VGAT-Venus* mouse GL. Circles show positions of added (yellow) and lost (pink) Venus+ cells. Added and lost neurons are indicated by yellow and pink arrows, respectively. RBC flow on Day 21 is shown in the right panel. (**P**) Density of newly added neurons in the perivascular region (paired *t*-test; seven mice). Data are presented as the means ± standard error of the mean (SEM). *$p < 0.05$, **$p < 0.01$. Scale bars: B, 30 μm; C, 40 μm; E, 1 μm; F, 10 μm; M, 20 μm; N, 20 μm; P, 10 μm. See also *Figure 1—figure supplements 1–3*.

The online version of this article includes the following video and figure supplement(s) for figure 1:

**Figure supplement 1.** Directional migration of new neurons relative to local blood flow.

**Figure supplement 2.** New neurons migrate along endomucin-negative vessels.

**Figure supplement 3.** New neurons exhibit a preference for arteriole-side vessels.

**Figure 1—video 1.** A three-dimensional image from the rostral migratory stream of a *Dcx-EGFP* (green) mouse infused with RITC-Dex-GMA (red).
https://elifesciences.org/articles/99502/figures#fig1video1

**Figure 1—video 2.** High-magnification view extracted from *Figure 1—video 1*.
https://elifesciences.org/articles/99502/figures#fig1video2

**Figure 1—video 3.** A three-dimensional image from the rostral migratory stream of a *Dcx-EGFP/Flt1-DsRed* mouse.
https://elifesciences.org/articles/99502/figures#fig1video3

**Figure 1—video 4.** A three-dimensional, high-magnification image of chain-forming new neurons in the rostral migratory stream.
https://elifesciences.org/articles/99502/figures#fig1video4

**Figure 1—video 5.** A three-dimensional image of immature neurons leaving the ventral ventricular–subventricular zone in a 1-month-old common marmoset infused with RITC-Dex-GMA (red).
https://elifesciences.org/articles/99502/figures#fig1video5

**Figure 1—video 6.** A three-dimensional image of an immature neuron from the ventral striatum in a 1-month-old common marmoset infused with RITC-Dex-GMA (red).
https://elifesciences.org/articles/99502/figures#fig1video6

cytoplasm with many free ribosomes, and an electron-dense nucleus with multiple nucleoli (*Doetsch et al., 1997*), to thin astrocytic endfeet enwrapping blood vessels (*Figure 1E*), as previously reported in the RMS and GCL (*Bovetti et al., 2007*; *Whitman et al., 2009*). These results indicate that new neurons migrate along blood vessels through their entire migration route, suggesting the possibility that their movement may be influenced by blood flow.

To examine the relationship between neuronal migration and blood flow, we recorded movement of new neurons and red blood cell (RBC) flow using two-photon laser scanning microscopy. Neuronal migration was recorded in the GL of *Dcx-EGFP* mice, in which new neurons were labeled with enhanced GFP (EGFP) (*Figure 1F*). RBC flow was also recorded in each vessel segment in the GL by line-scan measurements (*Figure 1F*). We identified migrating new neurons as EGFP+ cells whose position changed during the imaging period in *Dcx-EGFP* mice, in which the majority of EGFP+ cells are stationary. Most EGFP+ migrating cells (90.2%) were located within 10 μm of the nearest vessel (*Figure 1G*). To compare the blood vessel-guided migration of new neurons among blood vessels with different flows, we classified the vessels into two groups, high- and low-flow vessels, using the median RBC flow velocity as the criterion (*Figure 1H*). New neurons were more abundant near high-flow vessels than near low-flow vessels (*Figure 1H*). These cells migrated at a small angle to the longitudinal axis of blood vessels (*Figure 1—figure supplement 1*), indicating that new neurons use blood vessels as migration scaffolds. Next, we compared the migration speed of new neurons between the low- and high-flow perivascular region, the area within 10 μm from the nearest vessel. Migration speed, as measured by the migration distance of the cell soma, was significantly higher in the perivascular region of high-flow vessels than in that of low-flow vessels (*Figure 1I*). A previous study showed

that new neurons undergo discontinuous migration, including migratory and stationary periods in the GL (*Liang et al., 2016*). The proportion of the migratory period of new neurons in the total imaging period was larger in high-flow perivascular regions than in low-flow regions (*Figure 1J*). Furthermore, the maximum migration speed, calculated from two consecutive imaging frames, was significantly higher for neurons migrating along high-flow vessels than for those migrating along low-flow vessels (*Figure 1K*). These results suggest that neuronal migration is promoted in perivascular regions with abundant blood flow.

To compare the distribution of new neurons in perivascular regions with different blood flows in deep brain regions, we used endomucin, which has been reported to be downregulated by shear stress on vascular endothelial cells in vitro (*Zahr et al., 2016*). Endomucin-negative vessels showed higher RBC flow than endomucin-positive vessels (*Figure 1—figure supplement 2A–C*), indicating that endomucin can be used as a marker for RBC flow velocity in vivo. The density of bromode-oxyuridine (BrdU)-labeled doublecortin (Dcx)+ new neurons was higher in the perivascular region of endomucin-negative vessels than in that of endomucin-positive vessels in the GL, GCL, and RMS (*Figure 1—figure supplement 2D–G*), suggesting that new neurons migrate along vessels with faster blood flow throughout their migration route.

We next compared the distribution of new neurons near arteriole- and venule-side vessels identified using SLC16A1, which is expressed in venules and venule-side capillaries (*Vanlandewijck et al., 2018*). The density of perivascular Dcx+/BrdU+ new neurons was significantly higher in SLC16A1-negative perivascular areas than in SLC16A1-positive areas (*Figure 1—figure supplement 3A, B*), indicating that they actively migrate in the vicinity of arteriole-side vessels (*Figure 1—figure supplement 3C*). To further examine the localization of perivascular new neurons, we analyzed the cell distribution in *NG2-DsRed* mice (*Zhu et al., 2008*), in which we could identify arterioles, venules, and capillaries based on the morphological differences in NG2+ mural cells as performed previously (*Hartmann et al., 2015*; *Hill et al., 2015*). Arterioles, capillaries, and venules were characterized by band-like smooth muscle cells, pericytes, and fenestrated-shaped smooth muscle cells, respectively (*Figure 1L*). A large proportion of Dcx+/BrdU+ new neurons was observed near capillaries in the GL (arterioles; 11.8 ± 2.39%, venules; 4.34 ± 0.212%, capillaries; 83.8 ± 2.25%) (*Figure 1M*). The frequency of perivascular neuronal migration was compared in arteriole- and venule-side capillaries (defined as capillaries one to three order branches away from the nearest arterioles and venules, respectively). The density of Dcx+/BrdU+ cells was higher near arteriole-side capillaries than that near venule-side capillaries (*Figure 1N*). Taken together, these data suggest that most neuronal migration occurs near capillaries and that new neurons prefer capillaries on the arteriole side to those on the venule side (*Figure 1—figure supplement 3C*).

Because blood vessel-guided neuronal migration in the adult brain is a conserved phenomenon across species (*Akter et al., 2020*; *Kishimoto et al., 2011*; *Shvedov et al., 2024*), we hypothesized that blood flow may also influence neuronal migration in other brain regions of primates. The neocortex, which supports higher-order brain functions and has undergone evolutionary expansion in primates, was selected as a target region. In common marmosets, but not in mice, V-SVZ-derived new neurons migrate toward the neocortex and ventral striatum (*Akter et al., 2020*; *Figure 1—videos 5 and 6*). In the ventral striatum of 3- to 4-month-old common marmosets, Dcx+ immature neurons localized more frequently near SLC16A1-negative vessels than near SLC16A1-positive vessels, regardless of migration modes (*Figure 1—figure supplement 3D, E*). A similar tendency was also observed in the neocortex (*Figure 1—figure supplement 3F*). These results suggest that immature neurons prefer to migrate along arteriole-side vessels in the common marmoset brain, and this phenomenon is common between rodents and primates.

Finally, we examined whether new neurons mature near high-flow vessels following their migration along these blood vessels. Neuronal maturation and blood flow were recorded in *VGAT-Venus* mice, in which GABAergic neurons are labeled with Venus. Maturation of new neurons was confirmed by observation of the same regions with an interval of 21 days (*Figure 1O*). The density of newly added neurons was significantly higher near high-flow vessels than near low-flow vessels (*Figure 1P*). Consistently, BrdU-labeled mature neurons were frequently observed in the perivascular region of endomucin-negative vessels (*Figure 1—figure supplement 2H*). These results suggest that blood vessel-guided neuronal migration supplies new neurons in regions of high blood flow.

## Decreases in blood flow affect neuronal migration

To investigate the effects of blood flow on neuronal migration, we performed bilateral carotid artery stenosis (BCAS), which decreases blood flow in the anterior portion of the brain (*Hattori et al., 2016*; *Shibata et al., 2004*). Lentiviruses expressing Venus were injected into the V-SVZ to label new neurons migrating toward the OB (*Figure 2A*). The proportion of Dcx+/Venus+ cells in the OB was significantly lower in the BCAS group compared with the Sham-operated group at 5 days post injection (dpi) (*Figure 2B, C*), suggesting that blood flow reduction impairs the tangential migration of new neurons in the RMS. To evaluate whether the reduced number of new neurons in the OB following BCAS (*Figure 2B, C*) is solely due to impaired migration, we examined cell proliferation and survival in the V-SVZ and RMS. Specifically, we quantified the density of Ki67+ proliferating cells and cleaved caspase-3+ apoptotic cells in the sham and BCAS groups. BCAS significantly decreased cell proliferation and increased cell death in both the V-SVZ and RMS (*Figure 2—figure supplement 1*), suggesting that reduced neurogenesis and/or survival may contribute to the decreased distribution of new neurons in the OB. To further examine the influence of blood flow changes on neuronal migration, we induced photothrombotic clot formation to reduce blood flow in individual vessels (*Figure 2D*). Migration of EGFP+ cells along vessels in *Dcx-EGFP* mice was recorded using two-photon imaging (*Figure 2E*). To prevent effects other than blood flow inhibition, clot formation was induced in upstream vessel fragments distant from vessels close to migrating neurons (*Figure 2D*). For clot formation, a restricted area of selected vessels was irradiated by a two-photon laser immediately after intravenous injection of rose bengal. Clot formation resulted in blood flow termination in downstream vessels (*Figure 2F*), which was followed by a decrease in the migration speed of EGFP+ new neurons along downstream vessels (*Figure 2E, H*). Laser irradiation without rose bengal did not affect the speed of neuronal migration (*Figure 2G*), indicating that the inhibition of neuronal migration was not due to laser irradiation. These data suggest that blood flow facilitates neuronal migration in the RMS and OB and that blood contains factors influencing neuronal migration.

## Ghrelin increases neuronal migration speed by promoting somal translocation

We focused on ghrelin, which can be delivered from the bloodstream to the brain parenchyma, including the OB tissue, by transcytosis across vascular walls (*Rhea et al., 2018*). In addition, a previous study showed that newly generated neurons expressed growth hormone secretagogue receptor 1a (GHSR1a), a ghrelin receptor, in the V-SVZ, RMS, and OB (*Li et al., 2014*). Consistent with this report, we detected *Ghsr1a* mRNA in Dcx+ new neurons in the V-SVZ, RMS, and OB (*Figure 3—figure supplement 1A*). A previous study showed that migration of V-SVZ-derived new neurons is attenuated in ghrelin knockout mice (*Li et al., 2014*), suggesting that ghrelin stimulates neuronal migration. At first, to examine transcytosis of ghrelin in the OB, we introduced fluorescently labeled ghrelin into the bloodstream. We found an accumulation of fluorescent ghrelin in the RMS and OB as reported previously (*Rhea et al., 2018*; *Figure 3A, B*, *Figure 3—figure supplement 2*). Fluorescence signals were observed in vascular endothelial cells and parenchymal tissue in the RMS and OB (*Figure 3B*), indicating that blood-derived ghrelin crosses the vascular wall into the brain parenchyma and is delivered to new neurons. In addition, we observed high levels of fluorescent signal in vascular endothelial cells of endomucin-negative, high-flow vessels (*Figure 3C, D*), which suggests that transcytosis of blood-derived ghrelin may occur more frequently in these vessels, potentially due to increased endothelial endocytosis. We found that some, but not all, vessels showed particularly strong fluorescent signals in parenchymal regions adjacent to the abluminal side of vascular endothelial cells, as visualized by CD31 immunostaining (*Feng et al., 2004*; *Figure 3A', A''*). To quantify this observation, we defined two regions of interest: Area I (perivascular area), within 10 µm of the abluminal surface of CD31-positive endothelium; and Area II (distant area), located 10–20 µm away (*Figure 3E*). Of note, Area I corresponds to the perivascular region where new neurons are frequently observed (*Figure 1*). To determine whether ghrelin transcytosis occurs more frequently in high-flow vessels, we quantified signal gradients in the extra-vessel regions as fold changes (Area I/Area II), as illustrated in *Figure 3E*. The proportion of vessel segments with >1.5-fold increases was significantly higher in endomucin-negative vessels than in endomucin-positive ones (*Figure 3F*). Furthermore, vessel segments with >2-fold increases were observed exclusively in the endomucin-negative group (6.48% ± 1.18%). These data suggest that, in high-flow vessels, blood-derived ghrelin accumulates more in the immediate

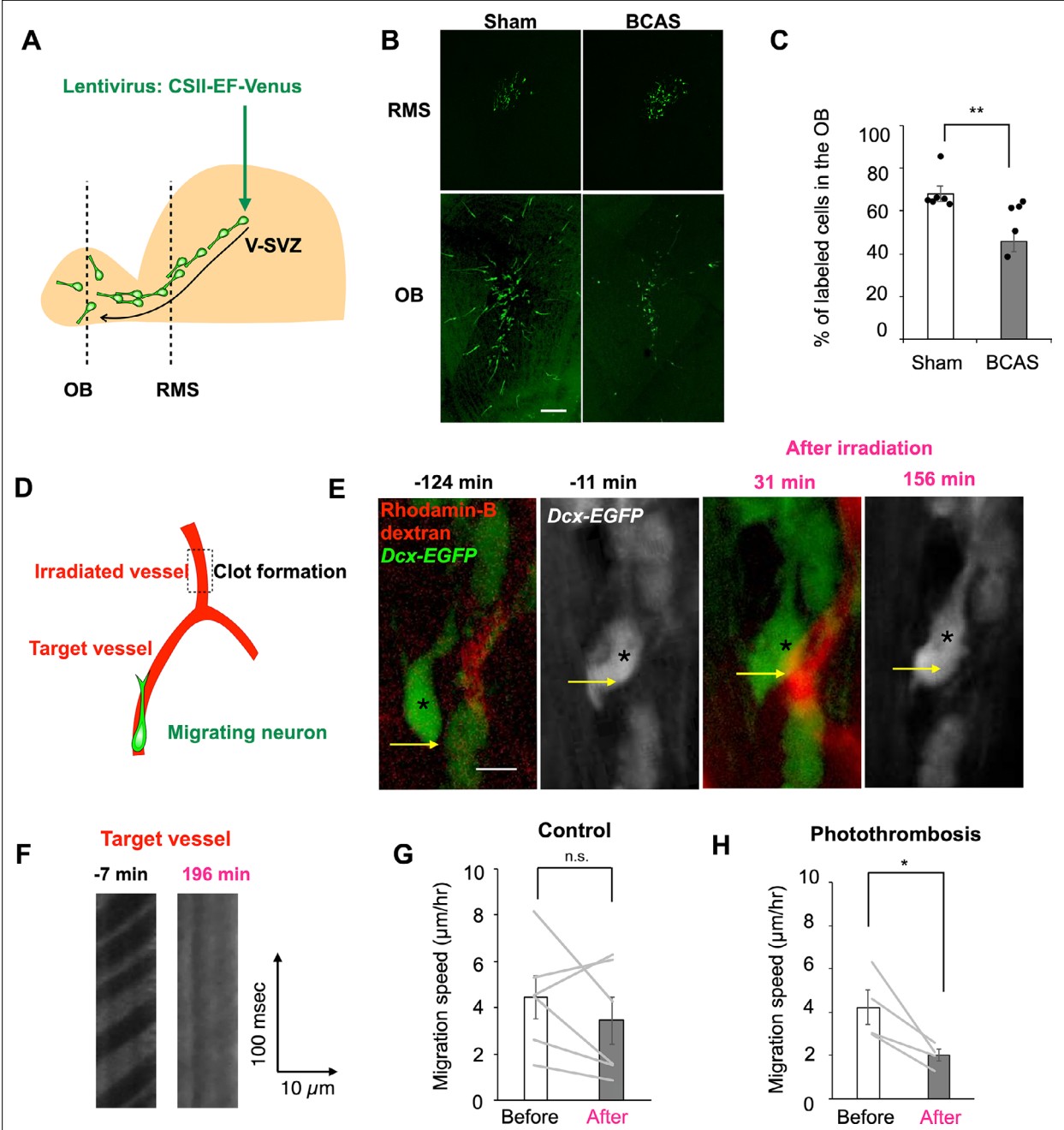

**Figure 2.** Blood flow inhibition attenuates neuronal migration. (**A, D**) Experimental schemes. (**B**) Fluorescent images of Venus+ new neurons (green) in the rostral migratory stream and olfactory bulb (OB). (**C**) Proportion of Venus+ cells in the OB in the Sham and bilateral carotid artery stenosis (BCAS) groups (Mann–Whitney *U*-test; Sham, six mice, BCAS, five mice). (**E**) Two-photon images of neuronal migration (arrows) before and after photothrombotic clot formation in a *Dcx-EGFP* mouse. A new neuron (green), a blood vessel (red). (**F**) Line-scan images from a blood vessel shown in (**E**). Comparison of migration speed before and after laser irradiation in the control (**G**) (paired *t*-test; six cells from six mice) and photothrombosis groups (**H**) (paired *t*-test; four cells from four mice). Data are presented as the means ± SEM. *$p < 0.05$, **$p < 0.01$, n.s., not significant. Scale bars: B, 100 μm; E, 10 μm.

The online version of this article includes the following figure supplement(s) for figure 2:

**Figure supplement 1.** Bilateral carotid artery stenosis (BCAS) affects cell proliferation and survival in the ventricular–subventricular zone (V-SVZ) and rostral migratory stream (RMS).

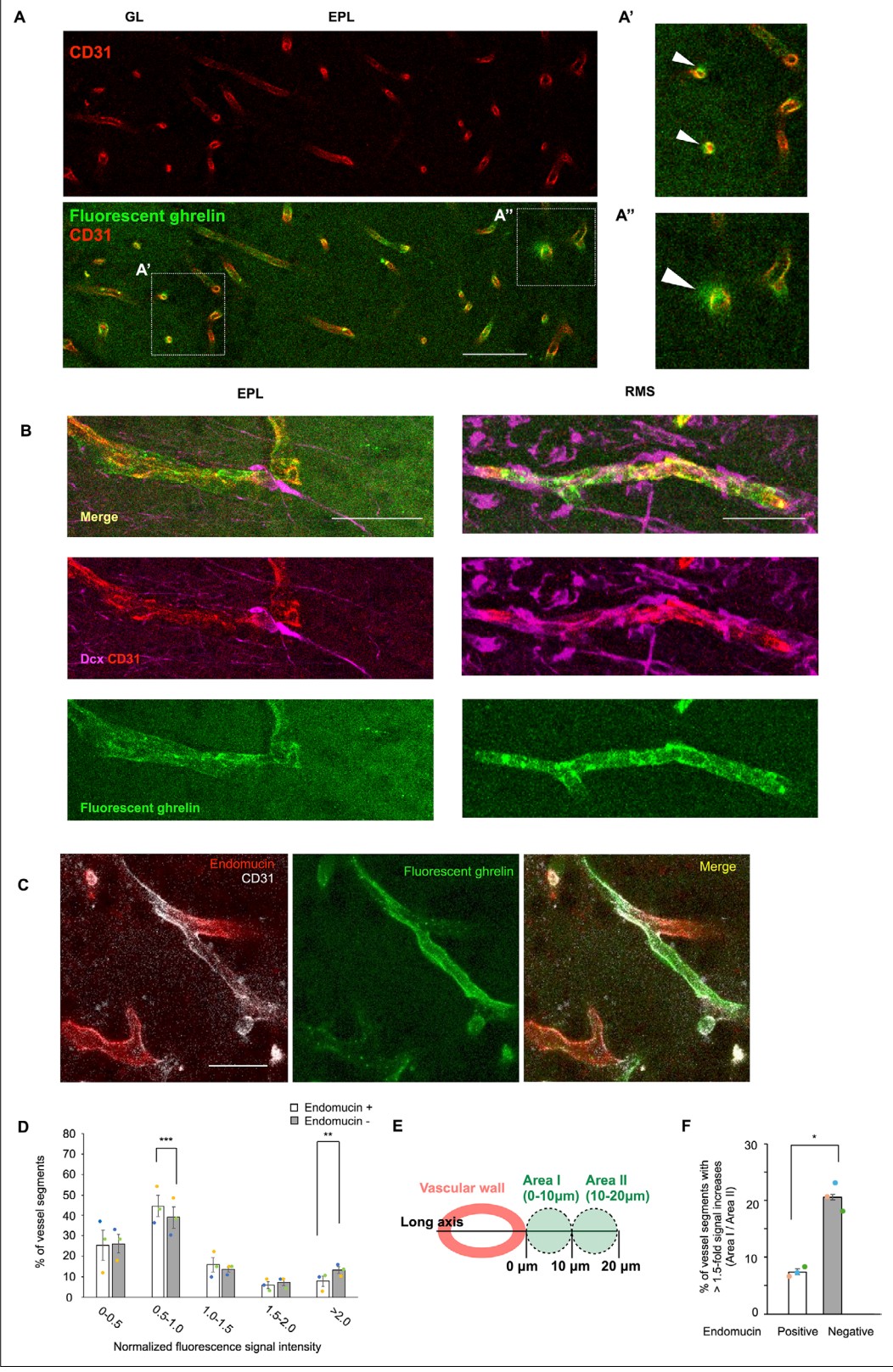

**Figure 3.** Ghrelin is delivered from the bloodstream to the rostral migratory stream (RMS) and olfactory bulb (OB) in the adult brain. (**A**) Representative images of the OB of a fluorescent ghrelin-infused mouse (6- to 12-week-old). CD31 (red), fluorescent ghrelin (green). (**A'**, **A''**) High-magnification images of boxed areas in (**A**). Arrowheads indicate fluorescent signals in parenchymal areas adjacent to the vascular endothelium. (**B**) Fluorescent images

*Figure 3 continued on next page*

*Figure 3 continued*

of neuronal migration along blood vessels in the external plexiform layer (EPL) and the RMS. CD31 (red), Dcx (magenta), fluorescent ghrelin (green). (**C**) Fluorescent images of blood vessels in the glomerular layer (GL). CD31 (white), endomucin (red), fluorescent ghrelin (green). (**D**) Normalized fluorescence signal intensity in vascular endothelial cells (paired *t*-test; three mice). (**E**) Schematic diagram for analyzing signal gradients in extra-vascular areas. (**F**) Percentage of vessel segments with >1.5-fold increases in Area I relative to Area II (paired *t*-test; three mice). Data are presented as the means ± SEM. *$p < 0.05$, **$p < 0.01$, ***$p < 0.005$. Scale bars: A, 50 µm; B, 20 µm (EPL), 10 µm (RMS); C, 20 µm. See also *Figure 3—figure supplements 1 and 2*.

The online version of this article includes the following figure supplement(s) for figure 3:

**Figure supplement 1.** New neurons express *Ghsr1a* mRNA in the adult brain.

**Figure supplement 2.** Blood-derived ghrelin enters the rostral migratory stream (RMS) and olfactory bulb (OB).

perivascular region than more distant areas. This supports the possibility that elevated blood flow increases the delivery of ghrelin to the vascular endothelium, enhancing its transcytosis into adjacent brain parenchyma. This mechanism may underlie the preferential migration of new neurons along high-flow perivascular regions, as shown in *Figure 1*.

To assess the direct effect of ghrelin on neuronal migration, we applied recombinant ghrelin to V-SVZ cultures, in which new neurons emerge and migrate as chains (*Figure 4A*). Ghrelin significantly increased the migration distance of these neurons (*Figure 4B*), indicating enhanced chain migration. We then used super-resolution time-lapse imaging to examine individually migrating neurons with or without knockdown (KD) of growth hormone secretagogue receptor 1a (GHSR1a), a ghrelin receptor expressed in V-SVZ-derived new neurons (*Li et al., 2014*; *Figure 4C*). Ghrelin enhanced the migration speed of control (*lacZ*-KD) cells, indicating that it also facilitates individual migration (*Figure 4D*). V-SVZ-derived new neurons exhibit saltatory migration consisting of a migratory phase and a resting phase (*Ota et al., 2014*). Ghrelin application increased the migratory phase proportion (*Figure 4E*) but not the length of the migration cycle (*Figure 4F*), suggesting that ghrelin signaling elongates the migratory phase of neuronal migration. Cultured new neurons alternate between leading process extension and somal translocation (*Figure 4C*). Ghrelin application did not affect the length or speed of leading process extensions (*Figure 4G, H*). In contrast, the somal translocation speed and somal stride length were significantly increased by ghrelin application (*Figure 4I, J*). No such effects were observed in *Ghsr1a*-KD cells (*Figure 4D, E, I, J*), suggesting that ghrelin promotes neuronal migration through ghsr1a. Taken together, these results suggest that ghrelin signaling promotes somal translocation and thus increases the efficacy of neuronal migration.

Ghrelin signaling has been reported to regulate actin cytoskeletal dynamics in astrocytoma cells (*Dixit et al., 2006*), which led us to examine whether a similar mechanism operates in migrating neurons. Somal translocation of migrating cortical interneurons is driven by formation of the actin cup, an accumulation of F-actin at the rear of the cell soma (*Martini and Valdeolmillos, 2010*). Time-lapse imaging of cultured new neurons expressing EGFP fused to the calponin homology domain of utrophin (EGFP-UtrCH), a fluorescent reporter for F-actin (*Burkel et al., 2007*), revealed discontinuous formation of the actin cup at the rear of the cell soma (*Figure 4K*). Ghrelin application extended the duration of actin cup formation (*Figure 4L*) and increased the migration distance during actin cup formation (*Figure 4M*, *Figure 4—video 1*). These effects were not observed in *Ghsr1a*-KD cells (*Figure 4L, M*, *Figure 4—video 2*), suggesting that GHSR1a-mediated ghrelin signaling promotes somal translocation of new neurons by activation of actin cup formation at the rear of the cell soma.

To investigate the effects of ghrelin signaling on neuronal migration in the OB in vivo, a mixture of lentiviruses encoding either *Ghsr1a*-KD or control shRNA was injected into the OB core of the same mouse (*Figure 5A*). Of the total labeled Dcx+ cells, the percentage of Dcx+ cells reaching the GL was significantly lower in the *Ghsr1a*-KD group than that in the control group (*Figure 5B, C*), suggesting that ghrelin enhances individual radial migration of new neurons in the OB. Coinjection of control and *Ghsr1a*-KD lentiviruses into the same site allowed us to directly compare their effects under identical conditions. However, this approach may allow reciprocal interactions between neurons infected with different constructs, potentially confounding cell-autonomous effects. To address this, we also performed separate injections of control and *Ghsr1a*-KD lentiviruses into different mice (*Figure 5—figure supplement 1A*). Consistent with the coinjection results, *Ghsr1a*-KD cells showed reduced distribution in the GL compared with that in control cells (*Figure 5—figure supplement 1B*), which

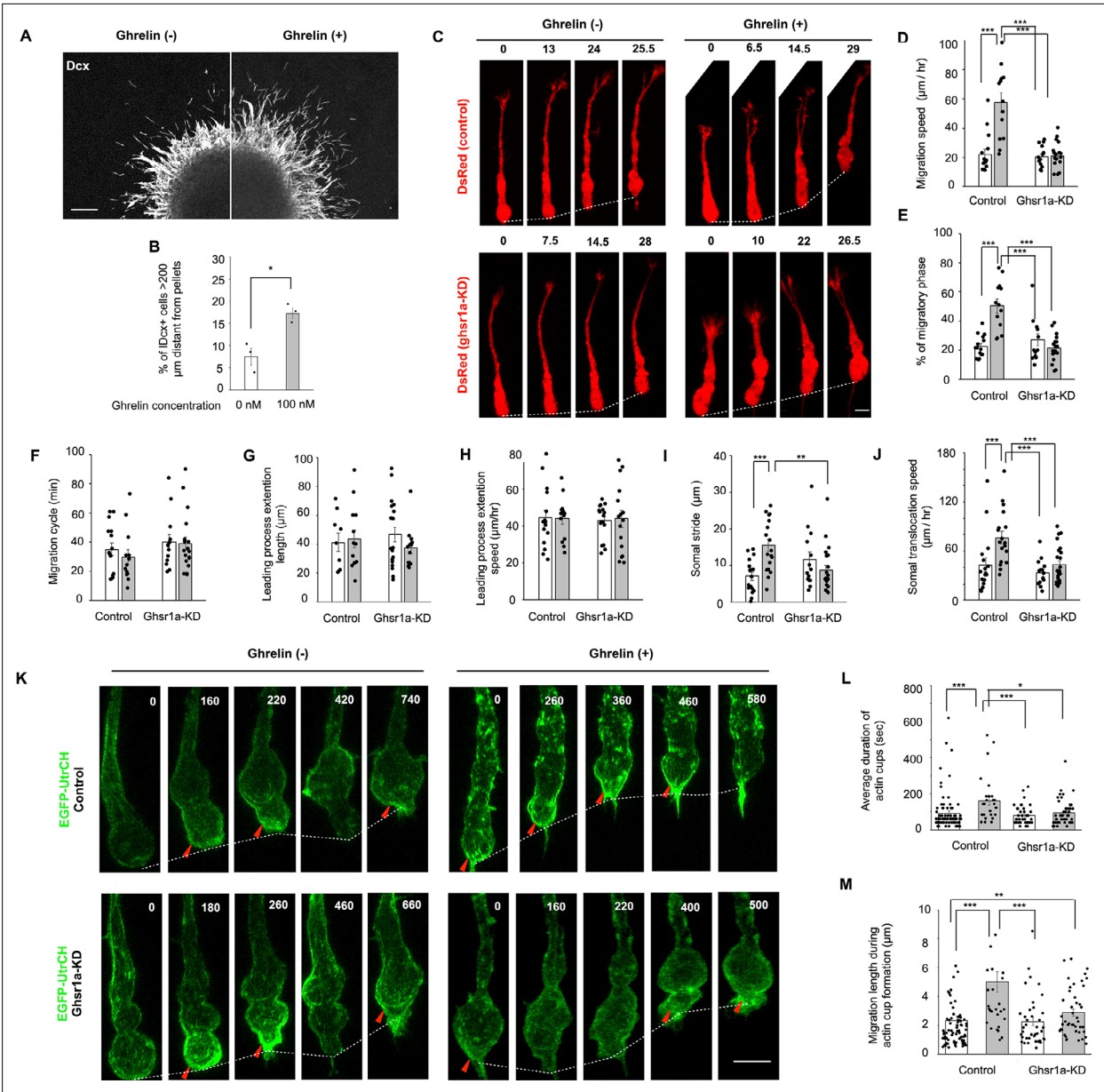

**Figure 4.** Ghrelin promotes neuronal migration by activation of actin cup formation. (**A**) Fluorescent images of Matrigel culture. Dcx (white). (**B**) Percentage of Dcx+ cells >200 µm distant from the edge of pellets (unpaired *t*-test; three independent cultures prepared on different days). (**C**) Time-lapse images of cultured new neurons expressing DsRed (red). The number above each panel indicates minutes after initiation of migration. (**D-J**) Migration speed (**D**), percentage of migratory phase (**E**), migration cycle (**F**), length/speed of leading process extension (**G, H**), and stride/speed of somal translocation (**I, J**) in neuronal migration (one-way ANOVA followed by Turkey–Kramer test; D–F; control/Ghrelin (−), 15 cells, control/Ghrelin (+), 13 cells, KD/Ghrelin (−), 13 cells, KD/Ghrelin (+), 18 cells, I, J; control/Ghrelin (−), 17 events, control/Ghrelin (+), 18 events, KD/Ghrelin (−), 14 events, KD/Ghrelin (+), 23 events). (**K**) Time-lapse images of actin cup formation (arrowheads) in the cell soma of new neurons. EGFP-UtrCH (green). Condensed dots of F-actin were scattered throughout the elongated cell soma in a control cell with ghrelin application. (**L, M**) Average duration of actin cups (**L**) and migration distance during actin cup formation (**M**) in new neurons (Kruskal–Wallis test followed by the Steel–Dwass test; control/Ghrelin (−), 79 cells, control/Ghrelin (+), 31 cells, KD/Ghrelin (−), 39 cells, KD/Ghrelin (+), 44 cells). Data are presented as the means ± SEM. *$p < 0.05$, **$p < 0.01$, ***$p < 0.005$. Scale bars: A, 100 µm; C, 5 µm; K, 5 µm.

The online version of this article includes the following video(s) for figure 4:

**Figure 4—video 1.** Actin cup imaging in control cells.
https://elifesciences.org/articles/99502/figures#fig4video1

**Figure 4—video 2.** Actin cup imaging in ghsr1a-knockdown cells.
https://elifesciences.org/articles/99502/figures#fig4video2

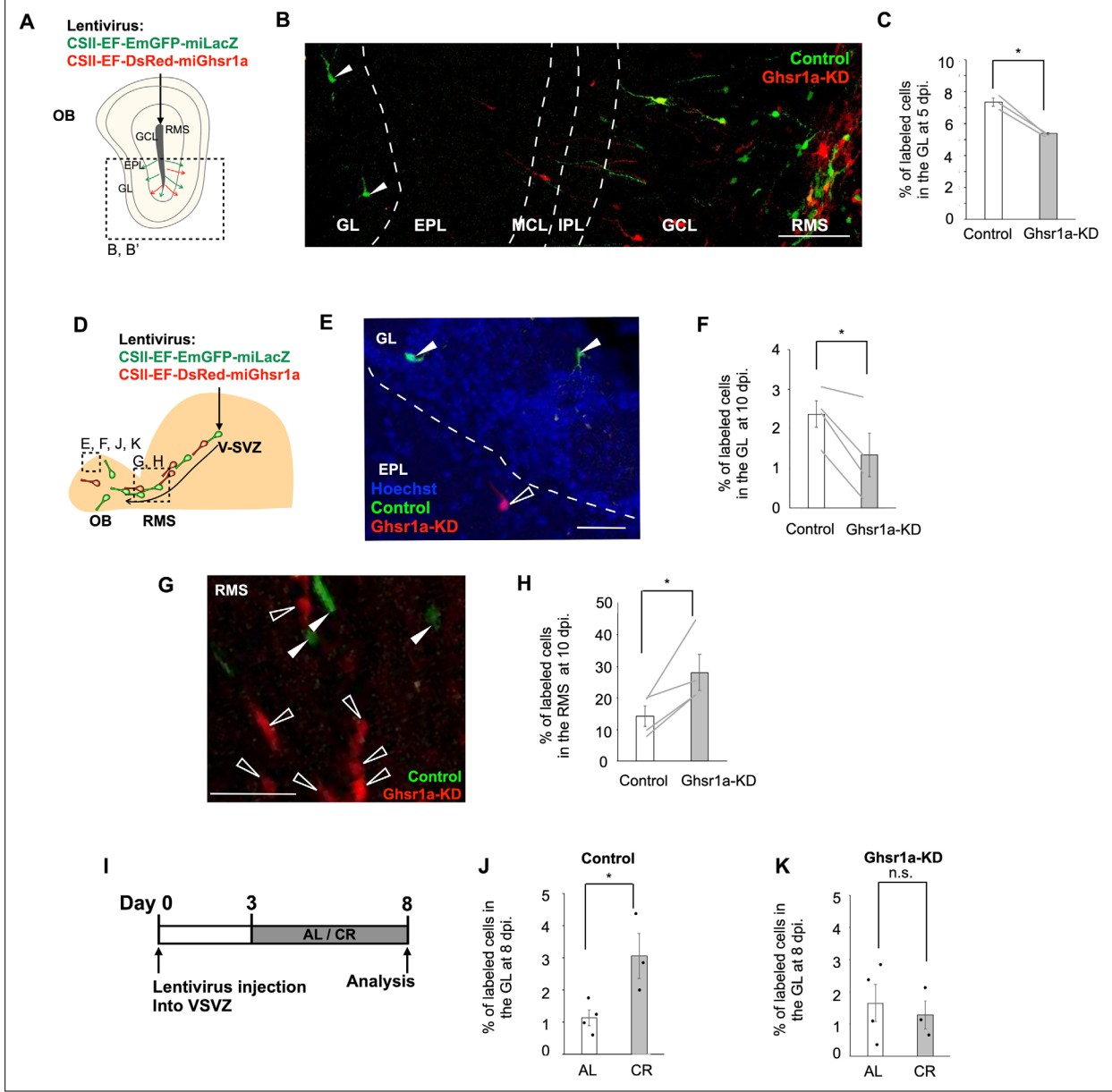

**Figure 5.** Ghrelin signaling promotes neuronal migration in the adult brain. (**A, D**) Experimental schemes. Lentivirus injection into the olfactory bulb (OB) core (**A**) and the ventricular–subventricular zone (V-SVZ) (**D**) was performed in 6- to 12-week-old adult mice. (**B**) Fluorescent images of new neurons in the OB in (**A**). (**C**) Proportion of labeled cells in the GL at 5 days post injection (dpi) in (**A**) (paired *t*-test; three mice). (**E, G**) Fluorescent images of new neurons in the GL (**E**) and rostral migratory stream (RMS) (**G**) for the experiments shown in (**D**). Control cells (white arrowheads), *Ghsr1a*-KD cells (clear arrowheads). (**F, H**) Proportion of labeled cells in the GL (**F**) and the RMS (**H**) at 10 dpi. in (**D**) (paired *t*-test; four mice). (**I**) Experimental scheme. (**J, K**) Proportion of labeled cells in the GL at 8 dpi in the ad libitum (**J**) and calorie restriction (**K**) groups (Control, unpaired *t*-test; AL, four mice, CR, three mice) (KD, unpaired *t*-test; AL, four mice, CR, three mice). Control (green), *Ghsr1a*-KD (red). GL (glomerular layer), EPL (external plexiform layer), MCL (mitral cell layer), IPL (internal plexiform layer), GCL (granule cell layer), RMS (rostral migratory stream), AL (ad libitum), CR (calorie restriction). Data are presented as the means ± SEM. \**p* < 0.05, n.s., not significant. Scale bars: B, 100 µm; E, 40 µm; G, 40 µm.

The online version of this article includes the following figure supplement(s) for figure 5:

**Figure supplement 1.** Ghrelin signaling is required for neuronal migration in the adult olfactory bulb (OB).

**Figure supplement 2.** *Ghsr1a*-KD does not affect cell proliferation of new neurons in the rostral migratory stream (RMS).

**Figure supplement 3.** *Ghsr1a*-KD does not affect the influence of bilateral carotid artery stenosis (BCAS) on neuronal migration.

**Figure supplement 4.** Calorie restriction (CR) promotes neuronal maturation in the olfactory bulb (OB).

supports that ghrelin signaling enhances neuronal migration in a cell-autonomous manner. When lentiviruses were injected into the V-SVZ (*Figure 5D*), *Ghsr1a*-KD (DsRed+) cells exhibited decreased distribution in the GL (*Figure 5E, F*) and increased distribution in the RMS compared with control (EmGFP+) cells (*Figure 5G, H*). These data indicate that ghrelin signaling facilitates both individual migration in the OB and chain migration in the RMS.

To determine whether the altered distribution of new neurons after *Ghsr1a*-KD is due to impaired migration rather than changes in cell production or survival, we assessed the effects of *Ghsr1a*-KD on the proliferation and survival of new neurons and their progenitors, which express GHSR1a (*Li et al., 2014*). We quantified the proportion of cleaved caspase-3+ cells and Ki67+ cells from the total labeled cells in the V-SVZ and RMS in both control and *Ghsr1a*-KD groups. We found no significant difference in cleaved caspase-3+ cell proportions between the groups (Control: 874 cells from five mice; *Ghsr1a*-KD: 678 cells from seven mice), suggesting that ghrelin signaling does not influence the survival of new neurons and their progenitors. Similarly, the percentage of Ki67 + cells in the RMS was similar between the two groups (*Figure 5—figure supplement 2*), indicating that *Ghsr1a*-KD does not impair cell proliferation in the RMS. However, technical limitations prevented a reliable evaluation of proliferation in the V-SVZ, as lentivirus injection into this region may interfere with GHSR1a expression in not only neural progenitors and new neurons, but also other ghsr1a-expressing cell types (*Zigman et al., 2006*). Although ghrelin signaling has been reported to promote cell proliferation in the V-SVZ (*Li et al., 2014*), our complementary in vitro KD experiments (*Figure 4C–J*) and in vivo OB-core lentivirus injections (*Figure 5A–C*), which did not affect the V-SVZ, consistently demonstrated that *Ghsr1a*-KD reduces neuronal migration. Taken together, our results suggest that blood-derived ghrelin enhances neuronal migration in the RMS and OB by stimulating actin cytoskeleton contraction in the cell soma, rather than by altering cell proliferation or survival.

As shown in *Figure 3*, ghrelin transcytosis preferentially occurs in high-flow blood vessels, indicating that ghrelin contributes to blood flow-dependent neuronal migration. However, the extent to which this process depends on ghrelin signaling remained unclear. To investigate this, we combined *Ghsr1a*-KD with BCAS. We found that BCAS reduced the percentage of *Ghsr1a*-KD new neurons reaching the OB, similar to the effect seen in control neurons (*Figure 5—figure supplement 3A, B*, see also *Figure 2A–C*). This suggests that blood flow influences neuronal migration even under *Ghsr1a*-KD conditions. Furthermore, we analyzed the distribution of *Ghsr1a*-KD neurons with respect to vessel flow characteristics. Even under *Ghsr1a*-KD, a higher proportion of new neurons were located in the area of endomucin-negative (high-flow) vessels compared with endomucin-positive (low-flow) vessels (*Figure 5—figure supplement 3C*), indicating that *Ghsr1a*-KD does not abolish the preferential association of migrating neurons with high-flow vessels. These findings suggest that ghrelin signaling is involved, but is not essential, for the blood flow-dependent guidance of migrating neurons, and that additional blood-derived signals may contribute to this process.

Finally, we examined whether CR, which has been reported to increase blood ghrelin levels (*Toshinai et al., 2001*; *Tschöp et al., 2000*), affects neuronal migration. We used a 70% CR protocol, which was previously reported to enhance hippocampal neurogenesis over 14 days (*Hornsby et al., 2016*). In our study, mice were fed 70% of their daily ad libitum (AL) food intake levels for 5 days (*Figure 5I*, *Figure 5—figure supplement 4A*). The proportion of labeled cells in the GL (*Figure 5J*) and NeuN+/Dcx− cells among BrdU+ cells in the OB (*Figure 5—figure supplement 4B, C*) was larger in the CR group than in the AL group. However, there was no significant difference in the proportion of *Ghsr1a*-KD cells in the GL between the control and CR groups (*Figure 5K*). We found no significant difference in the *Ghsr1a* expression level in OB Dcx+ cells between the AL and CR groups (*Figure 3—figure supplement 1B*). Taken together, these data suggest that blood flow promotes the migration of OB neurons during starvation via ghrelin signaling and that promoted migration of new neurons increases the number of mature neurons in the OB.

## Discussion

Previous studies have shown the role of blood vessels as physical scaffolds in the migratory routes of new neurons in various situations (*Bovetti et al., 2007*; *Fujioka et al., 2017*; *Grade et al., 2013*; *Kojima et al., 2010*; *Ohab et al., 2006*; *Snapyan et al., 2009*; *Sun et al., 2015*; *Thored et al., 2007*; *Whitman et al., 2009*; *Yamashita et al., 2006*). Proteins expressed by vascular cells have been reported to facilitate neuronal migration by binding to transmembrane receptors of new neurons

(*Fujioka et al., 2017*; *Grade et al., 2013*; *Ohab et al., 2006*; *Snapyan et al., 2009*). However, whether neuronal migration is affected by blood flow remains unknown. Therefore, in this study, we focused on the specific feature of the vasculature as a pipeline for blood delivery. The effects of blood flow on neuronal migration are difficult to detect with previously used experimental procedures such as immunohistochemistry of vascular endothelial cell markers in fixed tissue sections and time-lapse live imaging of brain slice cultures. Therefore, we classified blood vessels using molecular markers that reflect blood flow properties and performed in vivo live imaging to record blood flow. This approach enabled us to reveal the effects of blood flow on neuronal migration.

Previous studies reported that new neurons migrate along blood vessels in the RMS and GCL of the OB (*Bovetti et al., 2007*; *Snapyan et al., 2009*; *Whitman et al., 2009*). In the present study, three-dimensional imaging was performed over a wide area across the entire migration route in transparent brains, where the positional relationship between new neurons and blood vessels can be observed. The results suggest that new neurons use blood vessels as migration scaffolds throughout their migration route (*Figure 1A–D*). The distance we found between new neurons and blood vessels was larger than that reported in a previous study (*Snapyan et al., 2009*). This might be due to our method of measuring the distance from the blood vessels to the cell soma of new neurons, rather than the distance to the entire new neurons. To investigate the effects of blood flow on blood vessel-guided neuronal migration, we used two-photon imaging to record neuronal migration and blood flow in vivo in the GL in the superficial area of the OB, which is a useful model for analyzing blood vessel-guided neuronal migration. The results suggest that new neurons migrate faster near blood vessels with high flow than in the areas of those with low flow. Neural stem cells have been reported to increase blood flow in the adult V-SVZ (*Lacar et al., 2012*), raising the possibility that new neurons may increase blood flow in the OB. However, our observation that inhibition of blood flow suppressed neuronal migration suggests that blood flow facilitates neuronal migration. New neurons terminated their migration and differentiated into mature interneurons in perivascular regions with abundant flow. Thus, the blood flow-dependent mechanism of neuronal migration may supply new neurons to areas appropriate for their function.

We found that the migration speed of new neurons was decreased by blood flow reduction, suggesting that neuronal migration depends on blood flow in the adult brain. Although we cannot exclude the possibility that BCAS alters the cell proliferation or survival of new neurons, our photothrombotic clot formation experiments are better suited to directly examine how acute reductions in blood flow influence neuronal migration. These experiments allowed us to assess the migration speed of new neurons shortly after inducing localized blood flow inhibition. Based on the finding that neuronal migration was attenuated in our BCAS and photothrombosis experiments, we hypothesized that a blood-derived factor facilitates neuronal migration. A previous study demonstrated that the migration of V-SVZ-derived new neurons was attenuated in ghrelin knockout mice (*Li et al., 2014*). In our study, we found that the migration of cultured new neurons was enhanced by the application of ghrelin to the culture medium, and this effect was abolished by *Ghsr1a*-KD. These findings suggest that ghrelin directly stimulates neuronal migration through its receptor, GHSR1a, on new neurons. A previous study showed that GHSR1a is expressed in various regions of the brain (*Zigman et al., 2006*). In our experiments, new neuron-specific KD of *Ghsr1a* indicated that ghrelin signaling acts in a cell-autonomous manner to regulate neuronal migration. Nevertheless, a recent study (*Stark et al., 2024*) showed that GHSR1a was expressed in various cell types, including glutamatergic and GABAergic neurons, suggesting that ghrelin may also exert non-cell-autonomous effects on neuronal migration. Given the presence of diverse cell types—such as neurons, microglia, pericytes, and astrocytes—along the migratory route, it remains possible that GHSR1a activation in these neighboring cells contributes to the overall regulation of neuronal migration.

Furthermore, we identified the cellular and cytoskeletal mechanisms underlying this effect on migration. The results indicate that ghrelin enhances somal translocation during migration by activating actin cytoskeletal dynamics at the rear of the neuronal soma. We also found that ghrelin signaling increases the migration distance of cell soma, which increases the migratory phase duration. Further studies are needed to elucidate how ghrelin promotes actin cup formation in migrating neurons. Given that Rac, a Rho family GTPase, mediates actin remodeling downstream of ghrelin in astrocytoma cells (*Dixit et al., 2006*), it is possible that Rac may also be involved in ghrelin-induced cytoskeletal regulation in new neurons. In addition to actin remodeling, ghrelin may regulate microtubule dynamics. Ghrelin

signaling was shown to modulate microtubules via SFK-dependent phosphorylation of α-tubulin (*Slomiany and Slomiany, 2017*), raising the possibility that ghrelin promotes somal translocation of new neurons through coordinated regulation of both actin and microtubule networks (*Kaneko et al., 2017*). These results suggest that ghrelin signaling promotes neuronal migration in the RMS and OB in vivo, which could further strengthen our finding that blood flow plays a role in neuronal migration.

Furthermore, we found that blood-derived ghrelin crosses the vascular wall into the RMS and OB. Although we could not exclude the possibility that ghrelin is produced in the brain parenchyma (*Howick et al., 2017*), these results suggest that blood-derived ghrelin is provided to new neurons and promotes somal translocation by activating actin cup formation. Our results indicate that higher blood flow delivers a larger amount of ghrelin to the vascular endothelium, resulting in increased ghrelin transcytosis across vascular walls. It is possible that high blood flow increases the amount of ghrelin reaching the luminal surface of vascular endothelial cells, thereby increasing the possibility of ghrelin transcytosis into the brain parenchyma. Our findings suggest that blood-derived ghrelin contributes to the blood flow-dependent neuronal migration.

Because blood ghrelin levels increase during fasting (*Toshinai et al., 2001*; *Tschöp et al., 2000*), we examined the possibility that neurogenesis in the OB is affected by feeding conditions. We found that CR promoted the migration of OB neurons, an effect which was canceled by *Ghsr1a*-KD, suggesting that CR facilitates the OB neurogenesis through ghrelin signaling. A recent study demonstrated that fasting elevated *Ghsr1a* expression in the OB (*Stark et al., 2024*), raising the possibility that CR may have a similar effect. However, in our analysis, the number of *Ghsr1* mRNA puncta in Dcx+ new neurons did not differ between the AL and CR groups (*Figure 3—figure supplement 1B*), suggesting that CR does not alter GHSR1a expression levels in new neurons. Although we cannot exclude the possibility that CR modulates GHSR1a expression in other OB cell types, our combined CR and *Ghsr1a*-KD experiments support a cell-autonomous contribution of ghrelin signaling to CR-induced enhancement of neuronal migration. Although our data indicate that ghrelin signaling is essential for fasting-induced acceleration of neuronal migration, CR may also alter the concentrations of other circulating factors (*Alogaiel et al., 2025*; *Bonnet et al., 2020*; *Wu et al., 2024*), which could independently influence the behavior of migrating neurons. Since the supply of new neurons to the OB has been suggested to improve olfactory function in food-seeking behavior (*Lazarini and Lledo, 2011*), increased neurogenesis caused by long-term CR may contribute to improved olfactory function for food seeking during starvation.

The increased speed of somal translocation and elongated duration of the migratory phase in cultured new neurons are in accord with the in vivo increase in migration speed and duration of the migratory period of new neurons around high-flow vessels, respectively. Effective somal translocation has been suggested to be advantageous for cell migration in densely packed tissues (*Kengaku, 2018*). It is possible that the promotion of somal translocation by ghrelin signaling and blood flow overcomes difficulties in smooth migration of new neurons in dense tissues of the adult brain.

Our data showed that new neurons preferentially migrate along arteriole-side vessels rather than venule-side vessels in both mouse and common marmoset brains, suggesting that the mechanism of blood flow-dependent neuronal migration is conserved across rodent and primate species, as well as across brain regions. A previous study identified a ghrelin homolog in the common marmoset with high sequence similarity to the murine version (*Takemi et al., 2016*). In addition, the marmoset GHSR1a homolog shares 95.36% amino acid identity with that of the mouse (https://www.ncbi.nlm.nih.gov/protein/380748978). These findings suggest that blood-derived ghrelin promotes neuronal migration in the common marmoset brain in a manner similar to that in mice. Previous reports have shown that new neurons migrate along blood vessels to damaged areas after brain injury (*Fujioka et al., 2017*; *Grade et al., 2013*; *Kojima et al., 2010*; *Ohab et al., 2006*; *Thored et al., 2007*; *Yamashita et al., 2006*). Neuronal migration may be influenced by blood flow under pathological conditions as well as during blood vessel-guided migration under the physiological conditions shown in this study. It is possible that blood contains factors in addition to ghrelin that regulate neuronal migration. Blood flow may coordinate biological events between different organs by sending beneficial factors produced outside the brain to influence regionally restricted neuronal migration. Further studies could identify unknown factors involved in the mechanism of blood flow-dependent cell migration, which could contribute to the development of blood flow-based therapies for neurological diseases.

## Materials and methods

### Animals

All in vivo experiments were performed on 6- to 12-week-old C57BL/6J male mice. Wild-type mice were purchased from Japan SLC (Shizuoka, Japan). The following transgenic (Tg) mice were used: *Dcx-EGFP* BAC Tg mice (*Dcx-EGFP*) (*Gong et al., 2003*) provided by the Mutant Mouse Research Resource Center (MMRRC), *VGAT-Venus* BAC Tg line #39 (*VGAT-Venus*) (*Wang et al., 2009*), *Flt1-tdsDsRed* BAC Tg mice (*Flt1-DsRed*) (*Matsumoto et al., 2012*), and *NG2–DsRed* BAC Tg mice (*NG2-DsRed*) (*Zhu et al., 2008*). Cells were dissociated from postnatal day 0–1 (P0–P1) pups for in vitro experiments. Three- to four-month-old postnatal common marmosets were obtained from three mating pairs in a domestic animal colony and used for immunohistochemistry. All animals were housed under a 12-hr light/dark cycle with AL access to food and water. All experiments involving live animals were performed in accordance with the guidelines and regulations of Nagoya City University and the National Institute for Physiological Sciences.

### BrdU administration

BrdU (MilliporeSigma), dissolved in sterile phosphate-buffered saline (PBS), was intraperitoneally administered to mice (50 mg/kg) twice with an interval of 2 hr. Mice were fixed at 10 dpi to observe immature neurons or 28 dpi to observe mature olfactory neurons.

### Immunohistochemistry

Immunohistochemistry was performed as previously described for brain tissues of mice (*Sawada et al., 2011*) and common marmosets (*Akter et al., 2020*). Animals were transcardially perfused with PBS (pH 7.4) followed by 4% paraformaldehyde (PFA) in 0.1 M phosphate buffer (PB). The brains were removed from the skull and postfixed in the same fixative (24 hr for mice, 48 hr for common marmosets). Coronal sections were prepared using a vibratome (VT-1200S; Leica) (50 µm thick in mice, 60 µm thick in common marmosets). The sections were incubated with 10% normal donkey serum/0.2% Triton X-100 in PBS (blocking solution) for 30 min at room temperature (RT), the primary antibodies in blocking solution for 24 hr at 4°C, and AlexaFluor-/biotin-conjugated secondary antibodies (1:1000, Invitrogen) in the same solution for 2 hr at RT. For signal amplification, the sections were pretreated with 1% $H_2O_2$ in PBS for 40 min at RT before blocking. The signals were amplified with biotin-conjugated antibodies and a Vectastain Elite ABC Kit (Vector Laboratories) and visualized using Tyramide Signal Amplification (Thermo Fisher Scientific). For BrdU staining, sections were treated with 1 M HCl at 37°C for 30 min after 1% $H_2O_2$ treatment. After staining, the sections were mounted with aqueous mounting medium (PermaFluor, Lab Vision Corporation). Z-stack images were obtained using an LSM700 confocal laser scanning microscope (Carl Zeiss) with a 20× objective (512 × 512 pixels, 1.25 µm per pixel, 1 µm z-step size). For cell density analysis, the perivascular region was defined as the area within 10 µm of the edge of CD31+ vessels. BrdU administration was performed to identify relatively immature cells in the Dcx+ cell population including newly generated cells with different differentiation states.

The following primary antibodies were used: rat anti-GFP (1:500, 04404-84, Nacalai Tesque, Inc); rabbit anti-GFP (1:500, No. 598, Medical and Biological Laboratories Co, Ltd); rabbit anti-DsRed (1:2000, 632496, Clontech); guinea pig anti-doublecortin (Dcx) (1:400, ab2253, MilliporeSigma); rabbit anti-Dcx (1:500, #4604, Cell Signaling Technology); rat anti-CD31 (1:100, 550274, BD Biosciences); mouse anti-human CD31 (1:100, Dako); rat anti-endomucin (1:500, sc-65495, Santa Cruz Biotechnology); rat anti-BrdU (1:100, ab6326, Abcam); rabbit anti-NeuN (1:1000, ab177487, abcam); rabbit anti-SLC16A1 (1:500, TA321556, Origene); rat anti-Ki67 (1:500, #14-5698-82, eBioscience); and rabbit anti-cleaved caspase-3 (1:200, #9661, Cell Signaling Technology). Nuclei were stained with Hoechst (1:5000, H1399, Thermo Fisher Scientific).

### Three-dimensional imaging

For obtaining three-dimensional images from the RMS, new neurons were visualized in *Dcx-EGFP* or *Dcx-EGFP/Flt1-DsRed* mice. Because the population of EGFP+ cells includes not only V-SVZ-derived migrating new neurons but also other types of cells in *Dcx-EGFP*, new neurons were labeled with adenoviruses encoding enhanced GFP (Ad-GFP, Vector Biolabs) for the analysis of new neuron–blood vessel interactions. Adenoviruses were stereotaxically injected into the V-SVZ (1.0 mm anterior,

1.1 mm lateral to bregma, and 1.6–2.0 mm deep) to label new neurons in the RMS and injected into the RMS (2.8 mm anterior, 0.82 mm lateral to bregma, and 2.8–3.0 mm deep) to label new neurons in the OB. The blood vessel lumen was visualized with RITC-Dex-GMA as previously reported with modifications (*Miyawaki et al., 2020*). At 5 dpi, mice were transcardially perfused with PBS and 4% PFA/0.1 M PB followed by RITC-Dex-GMA. Mouse bodies were incubated in a 37°C water bath for 3 hr for polymerization. The brains were postfixed with SHIELD solutions (Lifecanvas Technologies) as previously reported (*Park et al., 2018*). Then, they were cleared using SmartClear II Pro (Lifecanvas Technologies). For visualization of new neurons in the common marmoset brain, the brains were incubated in 10% normal donkey serum/0.5% Triton X-100 in PBS for 30 min, the anti-Dcx primary antibody in blocking solution for 5 days, and the AlexaFluor-conjugated secondary antibodies (1:1000, Invitrogen) in the same solution for 3 days at 37°C. Refractive index matching was performed using EasyIndex (Lifecanvas Technologies) before imaging. Z-stack images were acquired with a light-sheet fluorescent microscope (Carl Zeiss) with a 5× and 20× objective (1216 × 1216 pixels, 1.3 μm per pixel, 1.2–1.4 μm z-step size). Three-dimensional reconstruction was performed using Imaris software (Oxford instruments) (https://imaris.oxinst.jp/). The three-dimensional distance was measured using ZEN software (Carl Zeiss) (https://www.zeiss.com/microscopy/ja/products/software/zeiss-zen.html) in light-sheet Z-stack images containing all of the GFP-positive cells in the OB hemisphere per mouse.

## Bilateral carotid artery stenosis

BCAS was performed as previously described with modifications (*Hattori et al., 2016*; *Shibata et al., 2004*). After midline incision of the mouse cervical region, microcoils with an inner diameter of 0.18 mm (Sawane Spring Co, Ltd) were wrapped around the common carotid arteries on both sides. Blood flow changes were confirmed by laser Doppler flowmetry in the anterior regions of brains (data not shown). A lentivirus encoding CSII-EF-Venus was stereotaxically injected into the V-SVZ to label new neurons. To analyze the cell distribution in the RMS and OB, mice were fixed at 5 dpi when glial activation is reported not to occur (*Shibata et al., 2004*).

## Two-photon imaging

As described previously (*Sawada et al., 2011*), thinned-skull surgery was performed on wild-type mice for comparisons between RBC flow and endomucin expression in identical vessels and on *Dcx-EGFP* and *VGAT-Venus* mice for observation of neuronal migration and maturation, respectively. Blood vessels were visualized by intravenous injection of Rhodamine-B dextran (D1841, Invitrogen) or Fluorescein dextran (D1823, Invitrogen). Mice were anesthetized by inhalation of isoflurane. The heads were immobilized with ear bars on a stereotactic stage before surgery. The skull over the OB was carefully thinned with a high-speed drill (MINITER Co, Ltd) and a surgical blade (Fine Science Tools). The thinned-skull window was observed under a two-photon laser scanning microscope (Nikon) and mode-locked system at 950 nm (Mai Tai HP, Spectra Physics) with a 25× water-immersion objective. Neuronal migration was recorded by identification of EGFP+ cells that changed their positions during the imaging period in the whole visible imaging field. During imaging, mice were anesthetized by intraperitoneal administration of a mixture of medetomidine (Meiji Seika Pharma Co, Ltd), midazolam (SANDOZ), and butorphanol (Meiji Seika Pharma Co, Ltd) (0.75, 4, and 5 mg/kg, respectively) and kept on a heating pad for maintenance of the body temperature at 37°C. Image stacks (2048 × 2048 pixels, 0.25 μm per pixel, 2 μm z-step size) were acquired every 30–60 min during 3 hr. RBC flow was recorded by serial line scans as previously reported (*Kleinfeld et al., 1998*). Line-shaped regions of interest were drawn along the longitudinal axis of each blood vessel. The RBC flow/s was calculated from repetitive scans obtained during 10 s at the beginning of the neuronal-migration recording. The median RBC flow velocity (38.4 RBCs/s) was used as a criterion for classification of vessels with different blood flows, whose distribution is not normal. Neuronal maturation was recorded as previously described with modifications (*Sawada et al., 2011*). After mice were anesthetized by isoflurane inhalation, the thinned-skull window was observed to record positions of Venus+ cells and RBC flow from each vessel in square-shaped regions (512 × 512 pixels, 0.5 μm per pixel, 2 μm z-step size). The same region of the GL was observed with an interval of 21 days. Stationary cells were defined as cells that were observed in the same position at both time points. Maturation and cell death in the GL were identified as addition and loss of Venus+ cells at the second time point. Data analysis was performed

using NIS Element software (Nikon) (https://www.microscope.healthcare.nikon.com/ja_JP/products/software/nis-elements).

## Photothrombotic clot formation

Photothrombotic clot formation was performed as previously reported with modifications (*Schaffer et al., 2006*). After identification of a blood vessel close to migrating neurons, an upstream vessel fragment was surrounded by a rectangular region of interest. Mice were intravenously injected with 20 mg/ml rose bengal (330000, Sigma-Aldrich) in PBS at concentration 0.05 mg/g. Immediately after injection, a selected fragment was irradiated using a two-photon laser at 950 nm. Irradiation by a 100-mW laser was performed for 5–10 s until the clot formed. The inner space of the vessel was equally irradiated by continuous movement of the imaging area. Ten minutes after the introduction of rose bengal, the RBC flow of a target vessel was recorded to confirm blood flow inhibition. The behavior of migrating neurons was observed for 3 hr after clot formation and compared with that before photothrombosis. As a control experiment, vessels without rose Bengal injection were irradiated with a two-photon laser. Samples were excluded if bleeding occurred or clots were dissolved during observation.

## Transmission electron microscopy

Transmission electron microscopy analysis was performed as previously described with modifications (*Matsumoto et al., 2019*; *Sawada et al., 2018*). Brain was fixed in 2.5% glutaraldehyde and 2% PFA in 0.1 M PB (pH 7.4) at 4°C, postfixed with 2% $OsO_4$ in the same buffer at 4°C, dehydrated in a graded ethanol series, placed in propylene oxide, and embedded in Durcupan resin for 72 hr at 60°C to ensure polymerization. Ultra-thin sections (60–70 nm) were cut using an ULTRACUT-E (Reichert-Jung) with a diamond knife, stained with 2% uranyl acetate in distilled water for 15 min, and stained with modified Sato's lead solution for 5 min. Sections were analyzed with a transmission electron microscope (JEM-1011J; JEOL, Tokyo, Japan). Migratory neurons were identified as cells with an elongated cell body, a dark cytoplasm with many free ribosomes, and an electron-dense nucleus with multiple nucleoli (*Doetsch et al., 1997*; *Matsumoto et al., 2019*; *Sawada et al., 2018*).

## Fluorescence in situ hybridization (RNA scope)

Fluorescence in situ hybridization was performed as previously described with modifications (*Miyamoto et al., 2024*) using the RNAscope multiplex assay (Advanced Cell Diagnostics), according to the manufacturer's protocol (*Wang et al., 2012*). After fixation with 4% PFA as described above, brains were postfixed for 4 hr at 4°C, incubated in 15% and 30% sucrose solutions in PBS, embedded in OCT compound (Sakura Finetek Japan), and frozen using dry ice and isopentane. Serial brain sections (10 μm) were cut using a cryostat (CryoStar NX70, Epredia). The sections were fixed again in 4% PFA at RT for 30 min, dehydrated in ethanol, treated with 1% hydrogen peroxide at RT for 30 min, and then boiled in RNAscope Target Retrieval Reagent (Advanced Cell Diagnostics) at 98°C for 15 min. Hybridization was performed using a probe targeting *Ghsr1a* mRNA (Mm-Ghsr-O2-C1, #241998, Advanced Cell Diagnostics) for 2 hr at 40°C. Following this, immunohistochemistry with rabbit anti-Dcx antibody and Hoechst staining was conducted.

## Protein labeling

For observation of ghrelin transcytosis across the vascular wall, recombinant octanoylated ghrelin (Human, sc-364689, Santa Cruz Biotechnology) was fluorescently labeled with Atto 647N NHS ester (18373, Sigma-Aldrich) as previously described with modifications (*Yang et al., 2020*). Ghrelin was dissolved in 0.1 M bicarbonate buffer (pH 8.3) at 2 mg/ml and reacted with Atto 647N NHS ester for 60 min at RT. After reactions, free label was removed by gel permeation chromatography with PD Mini-Trap G-25 columns (28918007, Cytiva). Mice were fixed as described above at 1 hr after intravenous injection of fluorescently labeled ghrelin (0.02 mg/30 g). Fluorescence signal intensity was measured by ZEN software. The average intensity among all vessels in each mouse was normalized to 1.0. To quantify signal gradients in extra-vessel regions, two circular ROIs (10 μm in diameter) were defined as illustrated in *Figure 3E*. The centers of the ROIs were aligned along the long axis of each vessel. Mean fluorescence intensities were measured in Area I (0–10 μm from the abluminal surface) and Area II (10–20 μm away), and fold changes (Area I/Area II) were calculated for each vessel segment.

## V-SVZ culture experiments

The V-SVZ was dissected from P0 to P1 mice, cut into blocks, and embedded in 60% Matrigel (BD Biosciences) in L-15 medium (Gibco). Cell aggregates were cultured in Neurobasal medium containing 2% NeuroBrew-21 (Invitrogen), 2 mM L-glutamine (Gibco), and 50 U/ml penicillin–streptomycin (Gibco) at 37°C in a 5% incubation system (Tokai Hit). For migration distance analysis, octanoylated ghrelin (Human, Rat, 1-10, Peptides International) was added to the medium at a final concentration of 100 nM at 24 hr post-embedding (hpe). Then, cell aggregates were fixed in 4% PFA/0.1 M PB at 48 hpe. For immunocytochemistry, aggregates were incubated in blocking solution for 30 min at RT, treated with the primary antibodies in blocking solution for 24 hr at 4°C, and treated with AlexaFluor-secondary antibodies (1:1000) in the same solution for 2 hr at RT. The migration distance was analyzed in three independent cultures prepared on different days.

## Viral vectors and plasmids

The pCSII lentiviral expression vectors were provided by Dr. Hiroyuki Miyoshi (RIKEN Tsukuba BioResource Center). The *lacZ*- and *Ghsr1a*-KD plasmids were generated as previously described (*Ota et al., 2014*; *Sawada et al., 2018*). The target sequences of *lacZ* mRNA and mouse *Ghsr1a* mRNA (Invitrogen) were inserted into modified Block-iT Poll II miR RNAi expression vectors containing EmGFP or DsRed-Express (Invitrogen). The Gateway System (Invitrogen) was used to generate pCSII-EF-Venus, pCSII-EF-Ghsr1a-IRES-Venus, pCSII-EF-EmGFP-express-miR-lacZ, pCSII-EF-DsRed-express-miR-Ghsr1a, pCAGGS-DsRed-express-miR-lacZ, and pCAGGS-DsRed-express-miR-Ghsr1a. For lentivirus production, the packaging vectors (pCAG-HIVgp, pCMV-VSV-G-RSV-Rev) and pCSII viral vectors were co-transfected into HEK-293T cells (RRID:CVCL_0063) to generate lentivirus particles. Then, the culture supernatants were concentrated by centrifugation at 8000 rpm for 16 hr at 4°C in an MX-307 refrigerated microcentrifuge (Tomy).

## *Ghsr1a*-KD experiments

The following sequence was inserted into siRNA expression vectors for targeting *Ghsr1a* mRNA (NM_177330.4): TGCTGAAGATGAGCAGATCGGAGAAGGTTTTGGCCACTGACTGACCTTCTCCGCTGCTCATCTTCAGG. For confirming efficacy of *Ghsr1a*-KD, pCSII-EF-Ghsr1a-IRES-Venus and/or pCSII-EF-DsRed-express-miR-Ghsr1a were co-transfected in HEK-293T cells. Venus signal was not observed in DsRed+ *Ghsr1a*-KD cells (data not shown).

For in vitro experiments, the dissected V-SVZ was dissociated with trypsin-EDTA (Invitrogen). The pCS2-EGFP-UtrCH was provided by Dr William Bement (University of Wisconsin-Madison) and Dr David J. Solecki (St. Jude Children's Research Hospital). The cells were washed in L-15 medium with 40 µg/ml DNaseI (Roche) and transfected with 2 µg plasmid DNA using the 4D-Nucleofector (Lonza). The cells were recovered in RPMI-1640 medium (Thermo Fisher Scientific) and embedded in 60% Matrigel in L-15. After cultivation in Neurobasal medium for 48 hr, time-lapse imaging was performed using an LSM880 confocal laser scanning microscope with a 40× objective (Carl Zeiss). Time-lapse images were captured at 30 s (*Figure 4C–J*) and 20 s (*Figure 4K–M*) intervals. The migration distance of cultured new neurons was measured using ImageJ manual tracking tools. A migratory phase was defined as a phase in which the cell soma traveled ≥30 µm during 1 hr, and a resting phase was defined as a phase in which the cell soma migrated <30 µm. ZEN software (Carl Zeiss) was used to analyze actin cup formation. Actin cups were defined as over 4-µm-long continuous EGFP-UrtCH signals that were 1.3 times brighter than those in other cell soma regions.

For in vivo experiments, the lentiviral solution was stereotaxically injected into the V-SVZ and OB core of adult mice (V-SVZ: 1.0 mm anterior, 1.1 mm lateral to bregma, and 1.6–2.0 mm deep) (OB core: 4.6 mm anterior, 0.9 mm lateral to bregma, and 0.6–0.9 mm deep). The proportions of EmGFP+/DsRed− and EmGFP−/DsRed+ cells among total labeled Dcx + new neurons in the RMS and OB were analyzed using ZEN software (Carl Zeiss). Double-positive cells (EmGFP+/DsRed+) were excluded from all analyses to avoid confounding effects of coinfection. CR was performed as previously reported (*Hornsby et al., 2016*). In migration analysis, calorie-restricted animals were fed 70% of the total food consumed by animals fed AL daily for the last 5 days prior to fixation at 8 dpi. In maturation analysis, CR was performed from Day 3 to Day 8 after BrdU administration at Day 0, following fixation at Day 15.

## Statistics

Sample sizes were not predetermined but were chosen based on previous studies. No specific strategy for randomization was employed, and no blinding was used. Statistical analysis was performed using EZR software (https://www.jichi.ac.jp/usr/hema/EZR/statmed.html) (*Kanda, 2013*). The normality of the data was analyzed using a Kolmogorov–Smirnov test or Shapiro–Wilk test. The equality of variance was analyzed using an *F* test. Comparisons of data between two groups were performed with unpaired/paired *t*-tests or Welch's *t*-test for normally distributed data and by Mann–Whitney *U*-tests/Wilcoxon signed-rank tests for abnormally distributed data. Comparisons among multiple groups were performed by one-way ANOVA/one-way repeated measures ANOVA/Kruskal–Wallis tests followed by a post hoc Tukey–Kramer test, Bonferroni test, or Steel–Dwass test. Numerical data are presented as the means ± standard error of the mean. A p value <0.05 was considered statistically significant. Significance is indicated in graphs as follows: *p < 0.05, **p < 0.01, ***p < 0.005, n.s., not significant.

## Acknowledgements

We thank M Agetsuma, K Eto, T Kobayashi, Y Yanagawa, S Nonaka, Y Uchida, R Mitsui, K Nishimura, H Takase, T Fujioka, T Miyamoto, the Laboratory Animal Facility and the Research Equipment Sharing Center at the Nagoya City University for technical support; W Bement, D J Solecki, H Miyoshi, and the MMRRC for materials; L Kreiner and L McCollum from Edanz and E Nakajima for editing a draft of this manuscript, and the Sawamoto Laboratory members for helpful discussions. This work was supported by research grants from the Japan Agency for Medical Research and Development (AMED) (24gm1210007, 25ym0126807 [to KS]), Japan Society for the Promotion of Science (JSPS) KAKENHI (25H01040, 25H02507, 24H02016, 24K22003, 23H04939, 20H05700, 19H04757, 18KK0213, 17H05750, JP22H04926, 26640046, 22122004 [to KS]), Core-to-core Program 'Neurogenesis Research & Innovation Center (NeuRIC)' (JPJSCCA20230007 [to KS]), Grant-in-Aid for Research at Nagoya City University (to KS), Cooperative Study Programs (22NIPS217) of the National Institute for Physiological Sciences (to KS), the Valencian Council for Innovation, Universities Science and Digital Society (PROMETEO/2019/075), the Spanish Ministry of Science, Innovation and Universities (PCI2018-093062), Grant-in-Aid for Promotion on Co-Creative Urban Development in Nagoya City University (2412145 [to KS]), Grant-in-Aid for Outstanding Research Group Support Program in Nagoya City University Grant Number (2401101 [to KS]), MEXT Project for promoting public utilization of advanced research infrastructure (JPMXS0441500024), the Mizutani Foundation for Glycoscience (to KS), and the Takeda Science Foundation (to KS).

## Additional information

### Funding

| Funder | Grant reference number | Author |
| --- | --- | --- |
| Japan Agency for Medical Research and Development | 24gm1210007 | Kazunobu Sawamoto |
| Japan Society for the Promotion of Science | 25H01040 | Kazunobu Sawamoto |
| Japan Society for the Promotion of Science | 26640046 | Kazunobu Sawamoto |
| Nagoya City University | 2412145 | Kazunobu Sawamoto |
| National Institute for Physiological Sciences | 22NIPS217 | Kazunobu Sawamoto |
| Valencian Council for Innovation, Universities Science and Digital Society | | José Manuel García-Verdugo |

| Funder | Grant reference number | Author |
| --- | --- | --- |
| Spanish Ministry of Science, Innovation and Universities | PCI2018-093062 | José Manuel García-Verdugo |
| Ministry of Education, Culture, Sports, Science and Technology | JPMXS0441500024 | Kazunobu Sawamoto |
| Mizutani Foundation for Glycoscience | | Kazunobu Sawamoto |
| Takeda Science Foundation | | Kazunobu Sawamoto |
| Japan Agency for Medical Research and Development | 25ym0126807 | Kazunobu Sawamoto |
| Japan Society for the Promotion of Science | 25H02507 | Kazunobu Sawamoto |
| Japan Society for the Promotion of Science | 24H02016 | Kazunobu Sawamoto |
| Japan Society for the Promotion of Science | 24K22003 | Kazunobu Sawamoto |
| Japan Society for the Promotion of Science | 23H04939 | Kazunobu Sawamoto |
| Japan Society for the Promotion of Science | 20H05700 | Kazunobu Sawamoto |
| Japan Society for the Promotion of Science | 19H04757 | Kazunobu Sawamoto |
| Japan Society for the Promotion of Science | 18KK0213 | Kazunobu Sawamoto |
| Japan Society for the Promotion of Science | 17H05750 | Kazunobu Sawamoto |
| Japan Society for the Promotion of Science | JP22H04926 | Kazunobu Sawamoto |
| Japan Society for the Promotion of Science | 22122004 | Kazunobu Sawamoto |
| Japan Society for the Promotion of Science | JPJSCCA20230007 | Kazunobu Sawamoto |
| Nagoya City University | 2401101 | Kazunobu Sawamoto |

The funders had no role in study design, data collection, and interpretation, or the decision to submit the work for publication.

## Author contributions

Takashi Ogino, Conceptualization, Formal analysis, Investigation, Writing – original draft, Writing – review and editing; Akari Saito, Formal analysis, Investigation, Writing – review and editing; Masato Sawada, Supervision, Investigation, Writing – review and editing; Shoko Takemura, Yuzuki Hara, Kanami Yoshimura, Investigation, Writing – review and editing; Jiro Nagase, Honomi Kawase, Takamasa Sato, Hiroyuki Inada, José Manuel García-Verdugo, Investigation; Vicente Herranz-Pérez, Investigation, Methodology; Yoh-suke Mukouyama, Resources, Investigation, Writing – review and editing; Masatsugu Ema, Resources; Junichi Nabekura, Supervision; Kazunobu Sawamoto, Conceptualization, Supervision, Funding acquisition, Writing – original draft, Project administration, Writing – review and editing

## Author ORCIDs

Takashi Ogino (ID) https://orcid.org/0000-0003-1619-8659
Shoko Takemura (ID) https://orcid.org/0000-0003-1872-4050
Vicente Herranz-Pérez (ID) https://orcid.org/0000-0002-1969-1214

Masatsugu Ema 
Junichi Nabekura 
Kazunobu Sawamoto 

### Ethics

All experimental procedures were approved by the guidelines and regulations of Nagoya City University (21-028, 21-030, 24-141, and 22-133) and the National Institute for Physiological Sciences (22A075, P08-044-A).

Reviewer #1 (Public review): https://doi.org/10.7554/eLife.99502.3.sa1
Author response https://doi.org/10.7554/eLife.99502.3.sa2

---

## Additional files

### Supplementary files

MDAR checklist

### Data availability

All data generated or analyzed during this study are included in the manuscript, supporting files, and the datasets in Dryad.

The following dataset was generated:

| Author(s) | Year | Dataset title | Dataset URL | Database and Identifier |
|---|---|---|---|---|
| Ogino T, Sawamoto K | 2025 | Data from: Neuronal migration depends on blood flow in the adult mammalian brain | https://doi.org/10.5061/dryad.dncjsxmcb | Dryad Digital Repository, 10.5061/dryad.dncjsxmcb |

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
