## [Editor Report · eLife Assessment]

This **fundamental** work provides novel insights into the blood flow-dependent mechanisms of neuronal migration and the role of Ghrelin signaling in the adult brain. The authors present **convincing** evidence that newborn rostral migratory stream (RMS) neurons are closely situated alongside blood vessels, preferentially along arterioles, and that migratory speed is correlated with blood flow. They also provide evidence (in vitro and some in vivo) that Ghrelin from blood is involved in augmenting RMS neuron migration speed.

---

## [Referee Report · Reviewer #1 (Public review)]

Summary:

This study provides compelling evidence suggesting that ghrelin, a molecule released in the surrounding of the major adult brain neurogenic niche (V-SVZ) by blood vessels with high blood flow controls the migration of newborn interneurons towards the olfactory bulbs.

Strengths:

This study is a tour de force as it provides a solid set of data obtained by time lapse recordings in vivo. The data demonstrate that the migration and guidance of newborn neurons relies on factors released by selective type of blood vessels.

Weaknesses:

Some intermediate conclusions are weak and may be reinforced by additional experiments.

Comments on revisions: The manuscript has improved.

---

## [Author Response]

The following is the authors’ response to the original reviews

**Reviewer #1 (Public review):**
Summary:This study provides compelling evidence suggesting that ghrelin, a molecule released in the surroundings of the major adult brain neurogenic niche (V-SVZ) by blood vessels with high blood flow, controls the migration of newborn interneurons towards the olfactory bulbs.Strengths:This study is a tour de force as it provides a solid set of data obtained by time-lapse recordings in vivo. The data demonstrate that the migration and guidance of newborn neurons rely on factors released by selective types of blood vessels.Weaknesses:Some intermediate conclusions are weak and may be reinforced by additional experiments.

We thank the reviewer for the thoughtful evaluation and constructive comments outlined in the “Recommendations for The Authors”. In response, we have incorporated additional data, revised relevant figures, and clarified explanations in the revised manuscript.

**Reviewer #2 (Public review)**
Summary:The authors establish a close spatial relationship between RMS neurons and blood vessels. They demonstrated that high blood flow was correlated with migratory speed. In vitro, they demonstrate that Ghrelin functions as a motogen that increases migratory speed through augmentation of actin cup formation. The authors proceed to demonstrate through the knockdown of the Ghrelin receptor that fewer RMS neurons reach the OB.They show the opposite is true when the animal is fasted.Strengths:Compelling evidence of close association of RMS neurons with blood vessels (tissue clearing 3D), preferentially arterioles. Good use of 2-photon imaging to demonstrate migratory speed and its correlation with blood flow. In vitro analysis of Ghrelin administration to cultured RMS neurons, actin visualization, Ghsr1KD, is solid and compelling.

We sincerely thank the reviewer for the encouraging comments and helpful suggestions. As noted, our original manuscript lacked sufficient in vivo evidence connecting blood flow with ghrelin signaling. To address this, we have added new data and revised the explanations throughout the manuscript as described below.

Weaknesses:(1) Novelty of findings attenuated due to prior work, especially Li et al., Experimental Neurology 2014. Here, the authors demonstrated that Ghrelin enhances migration in adultborn neurons in the SVZ and RMS.

We agree with the reviewer that the idea that ghrelin enhances migration of new neurons is not entirely novel. The study by Li et al. (2014) provided critical insights that guided our investigation into ghrelin as a blood-derived factor promoting neuronal migration. However, our study expands on this by demonstrating that ghrelin directly stimulates migration via GHSR1a in cultured new neurons, and we further identified the cellular and cytoskeletal mechanisms involved. Specifically, we showed that ghrelin enhances somal translocation by activating actin dynamics at the rear of the cell soma. We have revised the Results and Discussion sections accordingly to emphasize these novel aspects as follows:

“A previous study demonstrated that the migration of V-SVZ-derived new neurons was attenuated in ghrelin knockout mice (Li et al., 2014). In our study, we found that the migration of cultured new neurons was enhanced by the application of ghrelin to the culture medium, and this effect was abolished by Ghsr1a knockdown (KD). These findings suggest that ghrelin directly stimulates neuronal migration through its receptor, GHSR1a, on new neurons. A previous study showed that GHSR1a is expressed in various regions of the brain (Zigman et al., 2006). In our experiments, new neuron-specific KD of Ghsr1a indicated that ghrelin signaling acts in a cell-autonomous manner to regulate neuronal migration.” (Discussion, page 13, lines 10–18)

“Furthermore, we identified the cellular and cytoskeletal mechanisms underlying this effect on migration. The results indicate that ghrelin enhances somal translocation during migration by activating actin cytoskeletal dynamics at the rear of the neuronal soma.” (Discussion, page 13, lines 24–26)

(2) The evidence for blood delivery of Ghrelin is not very convincing. Fluorescently-labeled Ghrelin appears to be found throughout the brain parenchyma, irrespective of the distance from vessels. It is also not clear from the data whether there is a link between increased blood flow and Ghrelin delivery.

We agree that the correlation between blood flow and ghrelin transcytosis is not very convincing in our study. As the reviewer pointed out, Figure 3A gives the impression that fluorescent-labeled ghrelin is uniformly distributed throughout the brain parenchyma. However, high-magnification images newly added in Figure 3 show that some, but not all, vessels have particularly strong fluorescent signals in the parenchymal area adjacent to the abluminal side of vascular endothelial cells, visualized by CD31 immunostaining (Feng et al., 2004) (Figure 3A′, A′′). To quantify these observations, we defined two regions: Area I (perivascular area), within 10 μm of the abluminal surface of CD31-positive endothelium; and Area II (distant area), located 10–20 μm away (Figure 3E). Of note, Area I corresponds to the perivascular region where new neurons are frequently observed (Figure 1).

Importantly, we found strong ghrelin signals in vascular endothelial cells of endomucin-negative high-flow vessels (Figure 3C, D). This suggests that transcytosis of blood-derived ghrelin may occur more frequently in high-flow vessels due to increased endocytosis at the endothelium. To test this, we quantified signal gradients in the extra-vessel regions as fold changes (Area I / Area II), as illustrated in Figure 3E. The proportion of vessel segments with >1.5-fold increases was significantly higher in endomucin-negative vessels than in endomucin-positive ones (Figure 3F). Furthermore, vessels with >2-fold increases were observed exclusively in the endomucinnegative group (6.48% ± 1.18%).

These data suggest that, in high-flow vessels, blood-derived ghrelin accumulates more in the immediate perivascular region than in areas further away. This supports the possibility that elevated blood flow delivers a larger amount of ghrelin to the vascular endothelium, enhancing its transcytosis into adjacent brain parenchyma. This mechanism may underlie the preferential migration of new neurons along perivascular regions with high blood flow, as shown in Figure 1. We have incorporated this new data in Figure 3 and corresponding explanations into the Results, Figure legend and Methods

(3) The in vivo link between Ghsr1KD and migratory speed is not established. Given the strong work to open the study on blood flow and migratory speed and the in vitro evidence that migratory speed is augmented by Ghrelin, the paper would be much stronger with direct measurement of migration speed upon Ghsr1KD. Indeed, blood flow should also be measured in this experiment since it would address concerns in 2. If blood flow and ghrelin delivery are linked, one would expect that Ghsr1KD neurons would not exhibit increased migratory speed when associated with slow or fast blood flow vessels.

In Figure 3, we showed that ghrelin transcytosis occurs preferentially in high-flow vessels, suggesting a role for ghrelin in mediating the effects of blood flow on neuronal migration. However, whether this dependence is solely attributable to ghrelin signaling remains unclear.

To address this, we tested whether Ghsr1a-KD modifies the impact of reduced blood flow on neuronal migration by combining Ghsr1a-KD with bilateral common carotid artery stenosis (BCAS), a chronic cerebral hypoperfusion model (Figure S9A). We found that BCAS decreased the percentage of Ghsr1a-KD new neurons reaching the OB, similar to the effect seen in control neurons (Figure S9B, see also Figure 2A–C). This suggests that blood flow influences neuronal migration even under Ghsr1a-KD conditions.

Furthermore, we analyzed the distribution of Ghsr1a-KD neurons with respect to vessel flow characteristics. Even under Ghsr1a-KD, a higher proportion of new neurons were located in the area of endomucin-negative (high-flow) vessels compared with endomucin-positive (low-flow) vessels (Figure S9C), indicating that Ghsr1a-KD does not abolish the preferential association of migrating neurons with high flow vessels. These findings suggest that although ghrelin signaling contributes to blood flow-dependent migration, it is not the sole factor. Other blood-derived signals may also mediate this effect. We have included these new data in Figure S9 and updated the corresponding sections in the Results

**Reviewer #1 (Recommendations for the authors) :**
MajorPage 6, Line 13. Please provide in the result section some explanation about how photothrombic clot is induced.

We added the following explanation to the Results section to clarify the method used to induce photothrombotic clot formation.

“For clot formation, a restricted area of selected vessels was irradiated by a two-photon laser immediately after intravenous injection of rose bengal.” (Results, Page 7, lines 27–28)

Page 6, Line 18. The authors use the marmoset as an additional experimental model. Here, V-SVZ-derived newborn neurons migrate in other brain regions as compared to rodents. Please provide a clear rationale for moving from rodents to "common marmosets" as an experiment model. And why use marmosets only for this set of experiments?

We clarified the rationale for using common marmosets in addition to mice as follows:

“Because blood vessel-guided neuronal migration in the adult brain is a conserved phenomenon across species (Kishimoto et al., 2011; Akter et al., 2021; Shvedov et al., 2024), we hypothesized that blood flow may also influence neuronal migration in other brain regions of primates. The neocortex, which supports higher-order brain functions and has undergone evolutionary expansion in primates, was selected as a target region. In common marmosets, but not in mice, V-SVZ-derived new neurons migrate toward the neocortex and ventral striatum (Akter et al., 2021) (Supplemental Movies S4 and S5).” (Results, Page 6, lines 19–25)

Figure 2B. The experimental setup is possibly problematic as the lentiviral tracing measurement does not take into consideration the rate of neurogenesis or newborn neuron survival. Can authors assess the rate of proliferation and survival in the VSVZ/RMS upon BCAS to decipher whether the reduced number of cells observed in the OB only results from migration changes? (comparable remark stands for Figure 5)

To evaluate whether the reduction in the number of new neurons observed in the OB after BCAS (Figure 2B, C) is due solely to impaired migration, we assessed cell proliferation and survival in the V-SVZ and RMS. Specifically, we quantified the density of Ki67+ proliferating cells and cleaved caspase-3+ apoptotic cells in the sham and BCAS groups. BCAS significantly decreased cell proliferation and increased cell death in both the V-SVZ and RMS (Figure S4), suggesting that reduced neurogenesis and/or survival may contribute to the decreased neuronal distribution in the OB.

Although we cannot exclude the possibility that changes in cell proliferation or survival contributed to this effect, our photothrombotic clot formation experiments are better suited to directly examine how acute reduction in blood flow affects neuronal migration. These experiments allowed us to measure the migration speed of new neurons shortly after inducing localized blood flow inhibition. We found that clot formation significantly reduced the migration speed of new neurons (Figure 2E, H), indicating that blood flow changes directly impair neuronal migration in the adult brain.

We have included these new data in Figure S4 and updated the corresponding text in the Results, Discussion, Figure legend, and Methods as follows:

Figure 3. About ghrelin signaling. It is unclear whether its transcytosis occurs in endomucin-negative because of the high bloodstream flow. How can this be explained? What happens upon BCAS, is there still a close relation between ghrelin transcytosis, blood flow, and neuron migration?

As correctly noted, our initial explanation and data did not provide sufficient evidence that higher blood flow delivers a larger amount of ghrelin into the brain parenchyma. We found that some vessels had particularly strong fluorescent signals in the parenchymal area adjacent to the abluminal surface of vascular endothelial cells, as visualized by CD31 immunostaining (Feng et al., 2004) (Figure 3A′, A′′). On the basis of our observation that strong fluorescent signals were detected in vascular endothelial cells of endomucin-negative (high-flow) vessels (Figure 3C, D), we hypothesized that ghrelin transcytosis may occur more frequently in high-flow vessels due to increased endocytosis at the vessel endothelium.

To test this hypothesis, we quantified signal gradients in the extra-vessel regions by calculating fold changes in fluorescent intensity between two zones: Area I (0–10 μm from the abluminal surface of the endothelium) and Area II (10–20 μm away), as illustrated in Figure 3E. Area I corresponds to the perivascular region where new neurons are frequently found (Figure 1). We found that the proportion of vessel segments with >1.5-fold signal increase in Area I relative to Area II was significantly higher in endomucin-negative vessels than endomucin-positive ones (Figure 3F). Furthermore, vessel segments with >2-fold increases were observed exclusively in the endomucin-negative group (6.48% ± 1.18%). These results support the idea that higher blood flow increases the amount of ghrelin that reaches the luminal surface of vascular endothelial cells, thereby increasing the possibility of ghrelin transcytosis into the brain parenchyma.

We also examined whether blood flow inhibition–induced by BCAS or photothrombotic clot formation–affects the relationship between ghrelin transcytosis, blood flow, and neuronal migration. The above results suggest that blood flow reduction may decrease ghrelin transcytosis, thereby contributing to impaired neuronal migration. To further explore this, we analyzed the distribution of new neurons around high- versus low-flow vessels under BCAS conditions. In the BCAS group, we still observed a higher density of new neurons in the region of high-flow (endomucin-negative) vessels compared with in low-flow (endomucin-positive) ones (Figure S9C). This suggests that even under reduced blood flow, neuronal migration preferentially occurs near high-flow vessels. Taken together, these results suggest that ghrelin transcytosis, blood flow and neuronal migration are connected, and that this relationship persists under conditions of blood flow reduction.

Figure 4. Is ghrelin controlling both individual Dcx+ neuron migration as well as chain migration (cells moving more together)? This should be assessed and clarified.How is ghrelin controlling actin dynamics in newborn migrating neurons? Since somal translocation speed and somal stride length are both modulated by ghrelin, this factor may also control MT remodeling, could that be checked?

We have revised the manuscript to better explain the role of ghrelin in both modes of neuronal migration–chain and individual. Initially, we demonstrated that ghrelin enhances the migration of new neurons in V-SVZ culture (Figure 4A, B), where these neurons migrate outward as chains, indicating that ghrelin facilitates chain migration. In subsequent in vitro experiments (Figure 4C–M), we showed that ghrelin also enhances the migration of individual neurons. To examine this in vivo, we injected Ghsr1a-KD and control lentiviruses into two different anatomical regions: the V-SVZ, where chain migration originates, and the OB core, where new neurons migrate individually. These experiments enabled us to assess the role of ghrelin signaling in each mode of migration independently. We found that ghrelin enhanced both chain migration in the RMS and individual migration in the OB. These results indicate that ghrelin signaling facilitates both forms of neuronal migration. We added the following text in the Results section:

“To assess the direct effect of ghrelin on neuronal migration, we applied recombinant ghrelin to V-SVZ cultures, in which new neurons emerge and migrate as chains (Figure 4A). Ghrelin significantly increased the migration distance of these neurons (Figure 4B), indicating enhanced chain migration. We then used super-resolution time-lapse imaging to examine individually migrating neurons with or without knockdown (KD) of growth hormone secretagogue receptor 1a (GHSR1a), a ghrelin receptor expressed in V-SVZ-derived new neurons (Li et al., 2014) (Figure 4C). Ghrelin enhanced the migration speed of control cells (lacZ-KD) cells, indicating that it also facilitates individual migration (Figure 4D).” (Results, Page 9, lines 5–12)

“Of the total labeled Dcx+ cells, the percentage of Dcx+ cells reaching the GL was significantly lower in the Ghsr1a-KD group than in the control group (Figure 5B, C), suggesting that ghrelin enhances individual radial migration of new neurons in the OB.” (Results, Page 10, lines 5–8) “These data indicate that ghrelin signaling facilitates both individual migration in the OB and chain migration in the RMS.” (Results, Page 10, lines 17–18)

We also added discussion on how ghrelin may regulate cytoskeletal dynamics in migrating neurons. Ghrelin signaling has been reported to control actin cytoskeletal remodeling in astrocytoma cells (Dixit et al., 2006), which led us to investigate similar effects in migrating neurons. Rac, a member of the Rho GTPase family, was shown to mediate this actin remodeling in astrocytoma migration, suggesting it may also be involved in ghrelin-induced actin cup formation in new neurons. Furthermore, because somal translocation depends not only on actin but also on microtubule dynamics (Kaneko et al., 2017), it is possible that ghrelin influences both systems. Supporting this idea, ghrelin signaling was shown to modulate microtubule behavior via SFK-dependent phosphorylation of α-tubulin (Slomiany and Slomiany, 2017). These findings suggest that ghrelin may enhance somal translocation through coordinated regulation of both the actin and microtubule systems. We added following text in the Results and Discussion sections:

“Ghrelin signaling has been reported to regulate actin cytoskeletal dynamics in astrocytoma cells (Dixit et al., 2006), which led us to examine whether a similar mechanism operates in migrating neurons.”(Results, Page 9, lines 23–25)

“Further studies are needed to elucidate how ghrelin promotes actin cup formation in migrating neurons. Given that Rac, a Rho family GTPase, mediates actin remodeling downstream of ghrelin in astrocytoma cells (Dixit et al., 2006), it is possible that Rac may also be involved in ghrelininduced cytoskeletal regulation in new neurons.” (Discussion, Page 13, lines 28–31)

“In addition to actin remodeling, ghrelin may regulate microtubule dynamics. Ghrelin signaling was shown to modulate microtubules via SFK-dependent phosphorylation of α-tubulin (Slomiany and Slomiany, 2017), raising the possibility that ghrelin promotes somal translocation of new neurons through coordinated regulation of both actin and microtubule networks (Kaneko et al., 2017).” (Discussion, Page 13, line 31–Page 14, line 2)

It would also be informative to provide immunolabeling of Ghsr1 in the V-SVZ / RMS/ OB to have a clear picture of the expression pattern of this receptor. Newborn neurons migrate along blood vessels, which are surrounded by astrocytes that have also been reported to express Ghsr1, thus could newborn neuron migration change may also arise from activation of Ghsr1 in their surrounding astrocytes?

A previous study reported that GHSR1a is expressed in DCX+ new neurons in the RMS and OB, and in V-SVZ neural progenitor cells (Li et al., 2014). To visualize the spatial expression pattern of Ghsr1a, we performed RNAscope in situ hybridization because specific anti-GHSR1a antibodies suitable for immunohistochemistry were not available. Consistent with the previous report, we detected Ghsr1a mRNA in DCX+ new neurons in the VSVZ, RMS, and OB (Figure S5A), indicating that new neurons directly receive ghrelin signaling.

Moreover, our KD experiments demonstrated that ghrelin enhanced the migration of new neurons in a cell-autonomous manner via GHSR1a (Figure 4, 5). Nevertheless, a recent study (Stark et al., 2024) showed that GHSR1a was expressed in various cell types, including glutamatergic and GABAergic neurons, suggesting that ghrelin may also exert non-cellautonomous effects on neuronal migration. Given the presence of diverse cell types, including neurons, microglia, pericytes, and astrocytes, along the migratory route, it remains possible that GHSR1a activation in these neighboring cells contributes to the overall regulation of neuronal migration.

Figure 5. About the in vivo knockdown of Ghsr1a. The results section (page 9, line 3) mentioned that mice were either injected with one or the other construct but Figure 5 shows coincidence of GFP and dsRed positive cells. Were control and Ghsr1a shRNAs injected together into the same mouse? Could you quantify the number of cells in green (control), red (Ghsr1a KD), and yellow (both)? Won't they mostly be yellow? Have you tried injecting control and Ghsr1a separately? If yes, do you get the same result? Such analysis would be important to separate cell autonomous from noncell autonomous effects.

To minimize variability in injection conditions, we initially coinjected control and Ghsr1a-KD lentiviruses into the same mice and analyzed their migration using a paired design. As the reviewer correctly noted, some cells were coinfected and expressed both EmGFP and DsRed (18.7% ± 2.86% of EmGFP+ cells and 10.8% ± 0.533% of DsRed+ cells). To ensure that this overlap did not affect our analysis, we excluded EmGFP+/DsRed+ double-positive cells and focused solely on EmGFP+/DsRed− (control) and EmGFP−/DsRed+ (Ghsr1a-KD) single-positive cells.

We agree with the reviewer that coinjection could lead to reciprocal interactions between control and Ghsr1a-KD cells, potentially masking cell-autonomous effects. To address this, we performed an independent experiment in which control and Ghsr1a-KD lentiviruses were injected separately into different mice (Figure S7A), as suggested. Consistent with the results of the coinjection experiment, we found that the Ghsr1a-KD cells showed significantly reduced distribution in the GL compared with that in control cells (Figure S7B). Although we cannot exclude the possibility of a non-cell-autonomous effect of ghrelin, this result supports the conclusion that ghrelin signaling enhances neuronal migration in a cell-autonomous manner.

Who is expressing Ghsr1a, newborn neurons, and or their progenitors? The production and survival of newborn V-ZVS cells should be assessed upon knockdown of the ghrelin receptor too.

To determine whether the altered distribution of new neurons observed upon Ghsr1aKD is due to impaired migration rather than decreased cell production or survival, we examined the effects of Ghsr1a-KD on the proliferation and survival of new neurons and their progenitors, which express GHSR1a (Li et al., 2014).

We compared the proportion of cleaved caspase-3+ cells and Ki67+ cells from the total labeled cells in the V-SVZ and RMS between the control and Ghsr1a-KD groups. There was no significant difference in the proportion of cleaved caspase-3+ cells between the groups (Control: 874 cells from 5 mice; Ghsr1a-KD: 678 cells from 7 mice), suggesting that ghrelin signaling does not affect the survival of new neurons and their progenitors.

Similarly, the proportion of Ki67+ cells in the RMS did not differ significantly between the two groups (Figure S8), indicating that Ghsr1a-KD does not impair cell proliferation in the RMS. However, it remains technically difficult to evaluate whether Ghsr1a-KD affects proliferation in the VSVZ, because lentivirus injection into the VSVZ may interfere with GHSR1a expression not only in new neurons and neural progenitors, but also in other cell types known to express GHSR1a (Zigman et al., 2006). A previous study reported that ghrelin signaling promoted cell proliferation in the V-SVZ (Li et al., 2014), thus we cannot exclude the possibility that Ghsr1a-KD may affect V-SVZ proliferation.

To overcome this limitation, we assessed the effects of Ghsr1a-KD on neuronal migration using in vitro KD experiments (Figure 4C–J) and in vivo OB-core lentivirus injections (Figure 5A–C), both of which did not interfere with proliferation in the V-SVZ. These complementary approaches consistently demonstrated that Ghsr1a-KD reduces the migration speed of new neurons.

“To determine whether the altered distribution of new neurons after Ghsr1a-KD is due to impaired migration rather than changes in cell production or survival, we assessed the effects of Ghsr1aKD on the proliferation and survival of new neurons and their progenitors, which express GHSR1a (Li et al., 2014). We quantified the proportion of cleaved caspase-3+ cells and Ki67+ cells from the total labeled cells in the V-SVZ and RMS in both control and Ghsr1a-KD groups. We found no significant difference in cleaved caspase-3+ cell proportions between the groups (Control: 874 cells from 5 mice; Ghsr1a-KD: 678 cells from 7 mice), suggesting that ghrelin signaling does not influence the survival of new neurons and their progenitors. Similarly, the percentage of Ki67+ cells in the RMS was similar between the two groups (Figure S8), indicating that Ghsr1a-KD does not impair cell proliferation in the RMS. However, technical limitations prevented a reliable evaluation of proliferation in the V-SVZ, as lentivirus injection into this region may interfere with GHSR1a expression in not only neural progenitors and new neurons, but also other GHSR1aexpressing cell types (Zigman et al., 2006). Although ghrelin signaling has been reported to promote cell proliferation in the V-SVZ (Li et al., 2014), our complementary in vitro KD experiments (Figure 4C–J) and in vivo OB-core lentivirus injections (Figure 5A–C), which did not affect the V-SVZ, consistently demonstrated that Ghsr1a-KD reduces neuronal migration. Taken together, our results suggest that blood-derived ghrelin enhances neuronal migration in the RMS and OB by stimulating actin cytoskeleton contraction in the cell soma, rather than by altering cell proliferation or survival.” (Results, Page 10, line 19–Page 11, line 4)

“rat anti-Ki67 (1:500, #14-5698-82, eBioscience); and rabbit anti-cleaved caspase-3 (1:200, #9661, Cell Signaling Technology)” (Methods, Page 48, lines 14–16)

How much is ghrelin/Ghsr1 signaling conserved in marmosets?

How ghrelin signaling is conserved between mice and common marmosets is important to clarify. A previous study reported the existence of a ghrelin homolog in common marmoset, which shares high sequence similarity with that in mice (Takemi et al., 2016). Moreover, the GHSR1a homolog in the common marmoset (https://www.ncbi.nlm.nih.gov/protein/380748978) shares 95.36% amino acid identity with its mouse counterpart. These findings suggest that blood-derived ghrelin may similarly promote neuronal migration in the marmoset brain, as observed in mice.

We have added the following text in the Discussion section:

“Our data showed that new neurons preferentially migrate along arteriole-side vessels rather than venule-side vessels in both mouse and common marmoset brains, suggesting that the mechanism of blood flow-dependent neuronal migration is conserved across rodent and primate species, as well as across brain regions. A previous study identified a ghrelin homolog in the common marmoset with high sequence similarity to the murine version (Takemi et al., 2016). In addition, the marmoset GHSR1a homolog shares 95.36% amino acid identity with that of the mouse (https://www.ncbi.nlm.nih.gov/protein/380748978). These findings suggest that bloodderived ghrelin promotes neuronal migration in the common marmoset brain in a manner similar to that in mice.” (Discussion, Page 15, lines 8–16)

Page 9. Starvation has been shown to boost ghrelin blood levels. What is the exact protocol used in this experiment and is this indeed increasing Ghrelin release from blood vessels in the V-SVZ? What about Ghsr1 expression level in newborn neurons?

We have clarified the calorie restriction (CR) protocol used in our experiments. We adopted a 70% CR protocol, which was previously shown to enhance hippocampal neurogenesis when administered for 14 days (Hornsby et al., 2016). In our study, the daily food intake under ad libitum (AL) conditions was first measured, and CR mice were then fed 70% of that amount for 5 consecutive days (see Figure 5I and Figure S10A).

To assess whether CR enhances ghrelin transcytosis into the brain parenchyma, we performed ELISA to quantify ghrelin levels in the OB and RMS. However, ghrelin concentrations were below the detection limit in both groups, precluding a direct comparison.

We also considered whether CR modulates the expression level of the ghrelin receptor GHSR1a. A recent study reported that fasting increased GHSR1a expression in the OB (Stark et al., 2024), raising the possibility that CR may exert a similar effect. To test this, we performed in situ hybridization and quantified Ghsr1a mRNA puncta in Dcx+ cells in the OB. No significant difference was found between the AL and CR groups (Figure S5B), suggesting that CR does not alter GHSR1a expression levels in new neurons.

Although we cannot exclude the possibility that CR increases GHSR1a expression in other OB cell types, our combined CR and Ghsr1a-KD experiments strongly support a cellautonomous contribution of ghrelin signaling to the enhanced neuronal migration observed under CR conditions. Corresponding data and text have been added to Figure S5 and the Results, Discussion, and the Figure legend sections as follows:

MinorPage 4Line 19 In Supplemental movies 1 and 2, it is unclear where to see the GFP+ new neurons interact with BV. Can you add arrows as an indication for the readers? It will be better to add the anatomy term for orientation, caudal, or rostral in the video. (The same for Supplemental movies 3, 4, and 5).

To clarify the regions of interest in Supplemental Movies 1 and 2, where neuron–vessel interactions in the RMS are highlighted, we added dotted lines indicating the RMS boundaries. In addition, we created a new movie (Supplemental Movie S1′) showing a high-magnification view of Supplemental Movie S1, in which arrows mark EGFP+ new neurons interacting with blood vessels. We also added orientation indicators (e.g., caudal and rostral) and arrows to highlight new neuron–vessel interactions in Supplemental Movies S1–S5.

The following descriptions have been added to the Figure legends:

“Supplemental Movie S1′

High-magnification view extracted from Supplemental Movie S1. Arrows indicate EGFP+ cells interacting with blood vessels.” (Figure legend, Page 46, lines 6–8)

“Arrows indicate EGFP+ cells interacting with blood vessels.” (Figure legend, Supplemental Movie S3, Page 46, lines 16–17)

“Arrows indicate Dcx+ cells interacting with blood vessels.” (Figure legend, Supplemental Movies S4 and S5, Page 46, lines 21–22, 26–27)

Blood vessels are labeled in the Supplemental movies 2 and 3 by employing Flt1DsRed transgenic mice instead of RITC-Dex-GMA. However, Flt1-DsRed transgenic mice are not mentioned in the results section.

We have now included an explanation regarding the use of Flt1-DsRed mice, in which vascular endothelial cells were labeled with DsRed.

“To visualize blood vessels, we also used Flt1-DsRed transgenic mice, in which vascular endothelial cells were specifically labeled with DsRed (Matsumoto et al., 2012). Using DcxEGFP/Flt1-DsRed double transgenic mice, we observed close spatial relationships between new neurons and blood vessels (Supplemental Movies S2 and S3).” (Results, Page 4, lines 22– 26)

Figure 5. Can you indicate (in the figure legend and the result section) the stage of the adult brain used for this experiment?

We used 6- to 12-week-old adult male mice in all experiments in this study. To specify this, we have added the age of animals to both the Results and the relevant Figure legends as follows:

“Therefore, we first studied blood vessel-guided neuronal migration in the RMS and OB using three-dimensional imaging in 6- to 12-week-old adult mice, which enabled analysis of the in vivo spatial relationship between new neurons and blood vessels.” (Results, Page 4, lines 14–16)

“Figure 1 New neurons migrate along blood vessels with abundant flow in the adult brain.” (Figure legend, Page 25, line 4)

“(B, C) Three-dimensional reconstructed images of a new neuron (green) and blood vessels (red) in the rostral migratory stream (RMS) (B) and glomerular layer (GL) (C) of 6- to 12-weekold adult mice.” (Figure legend, Page 25, lines 6–8)

“(E) Transmission electron microscopy image of a new neuron (green) in close contact with a blood vessel (red) in the GL of a 6- to 12-week-old adult mouse.” (Figure legend, Page 26, lines 4–5)

“(F) Time-lapse images of a migrating neuron (indicated by asterisks) in the GL of a 6- to 12week-old Dcx-EGFP mouse.” (Figure legend, Page 26, lines 6–7)

“Figure 3 Ghrelin is delivered from the bloodstream to the RMS and OB in the adult brain (A) Representative images of the OB and cortex of a fluorescent ghrelin-infused mouse (6 to 12 weeks old).” (Figure legend, Page 30, lines 1–3)

“Lentivirus injection into the OB core (A) and the VSVZ (D) was performed in 6- to 12-week-old adult mice.” (Figure legend, Page 33, lines 3–4)

**Reviewer #2 (Recommendations for author):**
Major:Ghsr1KD and blood flow 2-photon experiments to directly measure migratory speed. Could also do the same with fasting with or without Ghsr1KD.

We thank the reviewer for the valuable suggestion to strengthen our study. As pointed out in the Public Review, we agree that direct in vivo measurement of neuronal migration speed under Ghsr1a-KD conditions is important to clarify the link between ghrelin signaling and blood flow.

Two-photon imaging is the most suitable method for this purpose. Although we attempted two-photon imaging of Ghsr1a-KD new neurons, the number of virus-infected cells observed in vivo was too low to yield reliable data. Therefore, we chose an alternative strategy, combining Ghsr1a-KD with blood flow reduction using the BCAS model (Figure S9A), in which migration speed can be quantified based on the percentage of labeled cells reaching the OB. As stated in the Public Review response, BCAS significantly decreased the migration speed of Ghsr1a-KD new neurons (Figure S9B), indicating that Ghsr1a-KD does not abolish the influence of blood flow reduction. These findings suggest that ghrelin signaling is involved, but is not essential, for blood flow-dependent neuronal migration.

As suggested by the reviewer, direct observation of migration dynamics (e.g., somal translocation, leading process extension, stationary and migratory phases) is needed, especially in calorie restriction experiments. Although our data indicate that ghrelin signaling is required for fasting-induced increases in migration speed of new neurons, calorie restriction could also change concentrations of other factors in blood (Bonnet et al., 2020; Wu et al., 2024; Alogaiel et al., 2025), which may independently affect behavior of migrating neurons. Given that ghrelin is not the sole factor contributing to blood flow-dependent neuronal migration, other circulating factors could affect behavior of migrating neurons in a different manner during fasting. In vivo twophoton imaging would be a powerful approach to determine whether fasting-induced neuronal migration is caused by upregulated somal translocation speed, which would further support a role for ghrelin in this process.

We have added the following text in the Discussion:

“Although our data indicate that ghrelin signaling is essential for fasting-induced acceleration of neuronal migration, calorie restriction may also alter the concentrations of other circulating factors (Bonnet et al., 2020; Wu et al., 2024; Alogaiel et al., 2025), which could independently influence the behavior of migrating neurons.” (Discussion, Page 14, lines 25–29)

Minor:(1) Show fluorescent Ghreliin in Figure 3 for all brain areas measured in Figure 1 (GL, EPL, GCL, and RMS) for direct comparison.

To allow for direct comparison across brain regions, we added a new Supplemental figure showing the distribution of fluorescently labeled ghrelin in the OB, including the GL, EPL, GCL and RMS. This comprehensive view highlights ghrelin localization relative to vasculature and migrating neurons in the regions analyzed in Figure 1.

(1) Figure 1, panel I is presented in a confusing manner. High blood flow points to 0 degrees, low blood flow to 180 degrees. It implies (unintentionally, I am sure) that low blood flow results in migration away from OB. Maybe plot separately?

We agree that the original presentation of Figure 1I could be misinterpreted as referring to anatomical orientation (i.e., toward or away from the OB). To avoid confusion, we revised the figure to categorize new neuron–vessel interactions into four groups according to (1) the angle between the migration direction and vessel axis (small or large), and (2) whether the new neuron is migrating toward or away from the direction of higher blood flow. This new presentation avoids implying a fixed anatomical direction and better reflects the relationship between local blood flow and neuronal migration behavior. The revised figure is presented as Supplemental Figure S1.